# A 900-year New England temperature reconstruction from *in situ* seasonally produced branched glycerol dialkyl glycerol tetraethers (brGDGTs)

Daniel R. Miller[1,2], M. Helen Habicht[2], Benjamin A. Keisling[2], Isla S. Castañeda[2], Raymond S. Bradley[1,2]

[1]Northeast Climate Adaptation Science Center, University of Massachusetts Amherst, Amherst, MA 01003, United States
[2]Climate System Research Center, Department of Geosciences, University of Massachusetts Amherst, Amherst, MA 01003, United States

*Correspondence to*: Daniel R. Miller (dmiller@geo.umass.edu)

**Abstract.** Paleotemperature reconstructions are essential for distinguishing anthropogenic climate change from natural variability. An emerging method in paleolimnology is the use of branched glycerol dialkyl glycerol tetraethers (brGDGTs) in sediments to reconstruct temperature but their application is hindered by a limited understanding of their sources, seasonal production, and transport. Here, we report seasonally resolved measurements of brGDGT production in the water column, in catchment soils, and in a sediment core from Basin Pond, a small, deep inland lake in Maine, USA. We find similar brGDGT distributions in both water column and lake sediment samples but the catchment soils have distinct brGDGT distributions suggesting that (1) brGDGTs are produced within the lake and (2) this in situ production dominates the downcore sedimentary signal. Seasonally, depth-resolved measurements indicate that most brGDGT production occurs in late fall, and at intermediate depths (18-30 meters) in the water column. We utilize these observations to help interpret a Basin Pond brGDGT-based temperature reconstruction spanning the past 900 years. This record exbibits similar trends to a pollen record from the same site and also to regional and global syntheses of terrestrial temperatures over the last millennium. However, the Basin Pond temperature record shows higher-frequency variability than has previously been captured by such an archive in the Northeastern United States, potentially attributed to the North Atlantic Oscillation and volcanic or solar activity. This first brGDGT- based multi-centennial paleo-reconstruction from this region contributes to our understanding of the production and fate of brGDGTs in lacustrine systems.

## 1 Introduction

Anthropogenic climate change is one of the most complex and challenging issues facing the world today and its impacts will likely be exacerbated in heavily populated areas, such as the Northeastern United States (NE US) (Fig. 1), a region comprised of communities that have been

historically susceptible to climate change (Horton et al., 2014). Here, over the past 120 years average temperatures have increased by ~1°C, precipitation has increased by 10%, and sea levels have risen by ~40 cm (Kunkel, 2013; NOAA, 2014). While historical records document the temperature increase of the past century, they are not long enough to capture the underlying variability of the pre-anthropogenic period. Therefore, high-resolution paleotemperature records,

such as those developed from lacustrine sedimentary sequences, are needed to investigate how current climate change compares to long-term natural variability. A regional synthesis of NE US late Holocene climate variability by Marlon et al. (2017) reviews temperature reconstructions from terrestrial sediment records using methods such as pollen (Gajewski, 1987; Webb et al., 2003; Oswald et al., 2007), testate amoeba (Clifford and Booth, 2013), and leaf wax hydrogen isotopic

ratios (Huang et al., 2004, Shuman et al., 2006; Gao et al., 2017). However, these climate proxies may also reflect changes in parameters other than temperature (i.e., precipitation, humidity, evapotranspiration, and vegetation) (Gajewski, 1988; Hou et al., 2008; Marlon et al., 2017). Therefore, additional quantitative paleotemperature records are needed to accurately assess past temperature variability in the NE US (Marlon et al., 2017).

Branched glycerol dialkyl glycerol tetraethers (brGDGTs), found globally in lakes, soils, rivers, and peats, provide an independent terrestrial paleothermometer well-suited to this task (e.g. Weijers et al., 2007; Peterse et al., 2012; De Jonge et al., 2013; Buckles et al., 2014; De Jonge et al., 2015). BrGDGTs are comprised of two ether-linked dialkyl chains containing zero to two methyl branches (prefixes I, II, and III) and zero to two cyclopentane moieties (suffixes a, b, and

c) (Sinninghe Damsté et al., 2000). Although the source organisms are unknown, these compounds are thought to be produced by Acidobacteria (e.g., Sinninghe Damsté et al., 2011; Sinninghe Damsté et al., 2018). Noting a strong correlation between mean annual air temperature (MAAT) and the degree of methylation of brGDGTs in global soils, Weijers et al. (2007) proposed that sedimentary brGDGTs could be used as a proxy for past soil temperature, which in many cases is

similar to mean annual air temperature. This motivated the development and later refinement of two indices, based on the degree of methlyation and cyclization of brGDGTs (MBT and CBT),

which were correlated to temperature and pH, respectively (e.g. Weijers et al., 2007; Peterse et al., 2012; De Jonge et al., 2014a).

More recently, improved chromatographic separation techniques for brGDGTs have been developed and indicated the presence of 5- and 6-methyl brGDGT isomers (De Jonge et al., 2013; De Jonge et al., 2014; Hopmans et al., 2016). The 6-methyl isomers may be abundant in environmental samples (De Jonge et al., 2014b), and failure to account for the presence of these compounds can have a significant influence on reconstructed temperatures (De Jonge et al., 2013; De Jonge et al., 2014a). Importantly, De Jonge et al. (2014a) demonstrated that soil pH (CBT) does not have an influence on the degree of methylation (MBT) and that earlier observations suggesting an influence of pH on methylation (Weijers et al., 2007) were the result of incomplete isomer separation. A new index based on the 5-methyl brGDGTs (MBT'$_{5ME}$) was developed and calibrated to temperature using a global soils dataset (De Jonge et al., 2014). MBT'$_{5ME}$ and 5-methyl fractional abundances have recently been calibrated for temperature reconstruction in lakes from East Africa (Russell et al., 2018) and China (Dang et al., 2018), respectively.

Initially, brGDGTs were presumed to be exclusively produced in soils, and subsequently washed into lakes or marine environments via erosion by rivers and streams (Hopmans et al., 2004). Further research demonstrated these compounds are also produced *in situ* in lakes and rivers (e.g. Tierney and Russell, 2009; Bechtel et al., 2010; Tierney et al., 2010; Zhu et al., 2011; Loomis et al., 2012; Schoon et al., 2013; Zell et al., 2013). Although some studies suggest that distinct brGDGTs are produced within the water column of lakes (Colcord et al., 2015; Weber et al., 2015) and show that their production is seasonally biased (e.g. Buckles et al., 2014; Loomis et al., 2014), relatively limited work has been done to understand their in-situ production and its consequences for the sedimentary brGDGT record (Zhang et al., 2016). Knowledge of brGDGT production and seasonality is important for appropriately calibrating and interpreting downcore records, yet few studies have combined modern observations of brGDGT distributions in the environment with a paleoclimate reconstruction for a temperate lake system.

Here we examine brGDGT abundances and distributions in catchment soil samples and at varying depths in the water column throughout the year at an inland lake in the NE US (Fig. 1). We collected samples from 2014–2015 to assess the seasonality and location of brGDGT production in and around Basin Pond, ME. We then use our observations to help interpret a 900 year-long

relative temperature record, providing the first decadally-resolved brGDGT-derived lacustrine paleoclimate reconstruction for this region.

## 2 Site information and field sampling

### 2.1 Study Site

Basin Pond, located in Fayette, ME (44º 28' N, 70º 03' W, elevation 124 m above sea level), is a small, deep lake with an area of 0.14 km$^2$ and a maximum depth of 32.6 m (Fig. 1). Basin Pond is fed from groundwater and precipitation, with one small, dammed, outlet stream running westward into the adjacent David Pond (Frost, 2005). Most of the 0.53 km$^2$ catchment area is dominated by a well-developed deciduous hardwood and evergreen forest, with only one residential building. Mean annual air temperature at Basin Pond is ~5.9°C and average annual precipitation is ~1150 mm (NOAA, 2014).

Basin Pond contains a unique sedimentary sequence comprised of annual laminations (varves) due to permanent water column stratification (Wetzel, 1983; Frost, 2005) resulting from a persistent thermocline. This stratification causes permanent bottom water anoxia, which enhances the preservation of annual laminations throughout the record (O'Sullivan, 1983). The Basin Pond varves are biogenic, with couplets comprised of a lighter, diatom-rich summer layer and a darker, humic winter layer (Frost, 2005).

The extent of anthropogenic impacts to Basin Pond and its catchment area have varied over the study interval. Although people were certainly present in Maine for the past 900 years, European-settler land clearance did not begin until the mid-1700s (Foster and Aber, 2004). It is uncertain whether the Basin Pond catchment was affected by this process. Due to its relatively remote location in New England, Maine experienced substantially less deforestation compared with the other NE US states (Foster and Aber, 2004). However, polycyclic aromatic hydrocarbons (PAHs) reflecting regional anthropogenic activity indicate that industrialization is notable in the Basin Pond sedimentary record (Miller et al., 2017). Furthermore, the lake's natural chemistry was disrupted in the 1950s, when Basin Pond was treated with a chemical piscicide, Rotenone (United States Geological Survey, 1996). Today, the lake is lightly used for recreation by members of the Basin-David-Tilton Ponds Association.

## 2.2 Sediment Coring

Sediment coring was performed from ice in March 2014, in the deepest part of the lake at 32 m (44º 27' 27" N, 70º 03' 09" W), using a UWITEC gravity coring system. Core BP2014-3D (52 cm) captured an undisturbed sediment-water interface and was subsampled in the field at 0.5 cm resolution for radioisotopic dating. Core BP2014-5D (174 cm) was immediately capped upon retrieval. Cores were split, photographed, and non-destructive down-core logging was performed using an Itrax XRF core scanner with a Molybdenum tube at 100 µm resolution in the Department of Geosciences at University of Massachusetts Amherst. Cores were kept refrigerated for one month prior to subsampling. Subsamples were stored frozen in WhirlPak bags until extraction.

## 2.3 Sediment trap construction, deployment and retrieval

Sediment traps were designed and constructed at University of Massachusetts Amherst. Sediment trap collection cones were made of high density polyethylene (HDPE) with a diameter of ~1 m (Fig. 1) and attached to 4L bottles for settling particulate matter (SPM) collection (Fig. 1). Note that our definition of SPM includes both material suspended in the water column and settling into the traps. Five sediment traps were deployed on May 27, 2014 at 6, 12, 18, 24, and 30 meter depth (Fig. 2). SPM was collected from all traps on 7/2/14, 8/16/14, 9/14/14 and 6/5/15. Each trap continuously accumulated SPM from deployment until collection and therefore each sample represents material collected over 36, 40, 28, and 264 days, respectively. The length of the last sampling period of 264 days was due to ice cover at the lake; sediment trap recovery was not possible until ice out. SPM labels on Figures 2, 3, & 5 and throughout the discussion are referred to by the month that was the midpoint of each collection period. Thus, the four sampling periods listed above are referred to respectively as June, July, September, and January SPM. Catchment soil samples were also collected around the perimeter of the lake at the time of initial trap deployment. All soil and water SPM samples were kept frozen until analysis.

## 3 Methods

### 3.1 Sedimentary Age Model

Subsamples for past climate reconstruction were taken every 0.5 cm from the uppermost 68 cm of core BP2014-5D. The age model for Basin Pond is based on $^{210}$Pb, varve counts, and five $^{14}$C dates

and was previously published by Miller et al., (2017). The sediment examined here ranges in age from modern to ~1100 BP, with a sampling resolution of 4 to 13 years (median: 7) (Miller et al., 2017).

### 3.2 Laboratory Methods

Overall, 10 catchment soil, 19 SPM sediment trap samples, and 136 sediment core samples were analyzed. Soil and lake sediment samples were freeze-dried and homogenized prior to extraction. For SPM samples, water from each collection bottle was filtered through a 47mm, 0.3-μm combusted Sterlitech glass fiber membrane filter, and dried prior to extraction. For most samples, a total lipid extract (TLE) was obtained using a Dionex Accelerated Solvent Extractor (ASE 200)

with a mixture of dichloromethane (DCM)/ methanol (MeOH) (9:1, v/v). For four SPM samples, plastic filters were washed and sonicated with HPLC-grade water, which was subsequently extracted with DCM three times. TLEs from SPM and catchment soil samples were separated into apolar (9:1 DCM/hexane v/v) and polar (1:1 DCM/MeOH v/v) fractions, while the lake sediment samples were separated into apolar, ketone (1:1 hexane/DCM) and polar fractions using alumina

oxide column chromatography. For all samples, one half of each polar fraction was filtered through 0.45μm PTFE syringe filters using 99:1 hexane/isopropanol (v/v). 0.1 μg of $C_{46}$ GDGT internal standard was added to each polar fraction prior to analysis. The other half of each polar fraction was derivatized using bistrimethylsiyltrifluoroacetamide (BSTFA), and algal biomarkers were identified with a Hewlett-Packard 6890 Series gas chromatograph coupled to an Agilent 5973 mass

spectrometer (GC-MS) using a Restek Rtx-5ms (60m x 250μm x 0.25 μm) column. Algal biomarkers, (iso)loliolide, $C_{30}$ 1, 13 *n*-alkyl diol, dinosterol/stanol, and β-sitosterol/stanol, were quantified with an Agilent 7890A dual gas chromatograph-flame ionization detector (GC-FID) equipped with two Agilent 7693 autosamplers and two identical columns (Agilent 19091J-416: 325 ᵒC: 60m x 320μm x 0.25 μm, HP-5 5% Phenyl Methyl Siloxan). For both the GC-MS and

GC-FID, helium was used as the carrier gas. The ovens began at a temperature of 70 ᵒC, increased at 10 ᵒC min $^{-1}$ to 130 ᵒC, increased again at 4 ᵒC min $^{-1}$ to 320 ᵒC, and then held for 10 minutes. The GCs were run in splitless mode. Compounds were quantified using an external calibration curve where squalane was injected at multiple concentrations ranging from 2 to 100 ng/μl; $r^2$ values for linearity tests were >0.99.

### 3.3 brGDGT analysis

Polar fractions were analyzed on an Agilent 1260 high performance liquid chromatograph (HPLC) coupled to an Agilent 6120 Quadrupole mass selective detector (MSD). Compound separation was achieved using the method of Hopmans et al. (2016). The technique uses two Waters UHPLC columns in series (150 mm × 2.1 mm × 1.7 μm) and isocratically elutes brGDGTs using a mixture of hexane (solvent A) and hexane: isopropanol (9:1, v:v, solvent B) in the following sequence: 18% B (25 minutes), linear increase to 35% B (25 minutes), linear increase to 100% B (30 minutes). Mass scanning was performed in selected ion monitoring (SIM) mode. brGDGTs were quantified with respect to the $C_{46}$ standard, assuming equal ionization efficiency for all compounds. For calculation of MBT'$_{5ME}$, CBT'$_{5ME}$, and the Isomer Ratio (IR), the following equations were used from De Jonge et al. (2014a&b):

$$\mathrm{MBT'}_{5ME} = \frac{\mathrm{Ia+Ib+Ic}}{\mathrm{IIIa+IIa+IIb+IIc+Ia+Ib+Ic}} \qquad \text{Eq. (1)}$$

$$\mathrm{CBT'}_{5ME} = -\log\left(\frac{\mathrm{Ib+IIb}}{\mathrm{Ia+IIa}}\right) \qquad \text{Eq. (2)}$$

$$\mathrm{IR} = \frac{\mathrm{IIa'+IIb'+IIc'+IIIa'+IIIb'+IIIc'}}{\mathrm{IIa+IIa'+IIb+IIb'+IIc+IIc'+IIIa+IIIa'+IIIb+IIIb'+IIIc+IIIc'}} \qquad \text{Eq. (3)}$$

For samples measured in duplicate (n=32), the maximum MBT'$_{5ME}$ difference was < 0.01, while the maximum CBT'$_{5ME}$ was 0.01; thus analytical error associated with proxy application is insignificant.

### 3.4 Time Series Analysis

To analyze the variance in the data presented here, we used the *Astrochron* R package (Meyers, 2012). Pre-processing of the data was kept to a minimum to avoid introducing spurious signals. The downcore brGDGT reconstruction was re-interpolated to 7-yr resolution (equivalent to median resolution of the raw data, see results) prior to spectral analysis. The published PAGES2k datasets were analyzed with their published chronologies, which is 1 year resolution for most regions and 10 years for the North American tree-ring based reconstruction (PAGES2k, 2013). Each of these reconstructions were smoothed to 7 year averages for easier comparison to our record.

## 4 Results

BrGDGTs were present in all soil, SPM and sediment core samples analyzed. In contrast, isoprenoid GDGTs, on which the $TEX_{86}$ temperature proxy is based (Schouten et al., 2002), were absent in a majority of samples or present in very low abundances compared to the brGDGTs. Therefore, $TEX_{86}$ could not be utilized as a temperature proxy at Basin Pond.

### 4.1 Catchment Soils

BrGDGTs Ia and IIa dominated distributions in the catchment soil samples, with relative abundances of 65% ±13% and 28% ±7%, respectively (Fig. 2). The next largest relative abundances were IIIa and Ib, comprising 3% ±6% and 3% ±1%, respectively (Fig. 2). Total brGDGT concentrations in soils ranged from 1.5 to 7.3 (median = 2.2) µg gsed$^{-1}$.

### 4.2 SPM

BrGDGTs Ia, IIa and IIIa dominated distributions in the SPM samples, with relative abundances of 28% ±8%, 37% ±7%, and 30% ±8%, respectively (Fig. 2). The next largest relative abundances were Ib and IIb, each comprising 2% ±<1%. In June and July 2014, group I brGDGTs were the most abundant, whereas in September 2014 and January 2015 reductions in group I brGDGTs were accompanied by increases in group III brGDGTs (Fig. 2). Overall, fluxes of brGDGTs were highest in September 2014 (ranging from 0.36 to 15.2 ng m$^2$ day$^{-1}$ at different depths) (Fig. 3). In June and July 2014, total brGDGT fluxes at various depths ranged from 0.009 to 0.04 ng m$^2$ day$^{-1}$ and 0.04 to 0.14 ng m$^2$ day$^{-1}$, respectively (Fig. 3).

BrGDGT fluxes and distributions also varied as a function of depth (Fig. 3,4, 5). In general, summed brGDGT fluxes increase with depth, with up to an order of magnitude higher fluxes at 30 m compared to 6 m for all dates (Fig. 3). BrGDGT fluxes in the upper and lower water column peaked in September with fluxes of 0.4 to 0.6 ng m$^2$ day$^{-1}$ and 7 to 16 ng m$^2$ day$^{-1}$, respectively. The average distributions also changed as a function of depth. Group I brGDGTs comprised 30% of the distributions at all depths. Group II brGDGTs had the greatest abundance at 12 and 18 m water depth (Fig. 5), comprising up to 50% of the distribution. Group III brGDGTs comprised an average of 30% with a peak of 35% at 30 m, and a minimum at 18 m of <20%. These distributions lead to variations in MBT'$_{5ME}$ and CBT'$_{5ME}$ indices as a function of depth (Fig. 5). MBT'$_{5ME}$ varied

from 0.34 to 0.46, and peaked at 24 m water depth (Fig. 4). CBT'$_{5ME}$ varied from 1.1 to 1.6, peaked at 12 m and then decreased with depth (Fig. 4).

### 4.3 Surface sediment samples

To represent surface sediments, we averaged the measurements from the uppermost 5 cm of Core BP2014-5D, corresponding to approximately 70 years (Miller et al., 2017). BrGDGTs Ia, IIa, and IIIa dominate the distribution, with relative abundances of 27%, 32%, and 33%, respectively (Fig. 3). The next largest relative abundances are IIb, Ib, and IIIb (3%, 2%, and 1%, respectively). MBT'$_{5ME}$ values in surface sediments range from 0.35 to 0.45 (Fig. 6).

### 4.4 Sediment Core

BrGDGT concentrations in these samples ranged from 1.2 to 21.1 (median: 8.09) µg gsed$^{-1}$. MBT'$_{5ME}$ ranged from 0.34 to 0.50 (median: 0.39). MBT'$_{5ME}$ values fluctuate around a stable mean from 1100-1400 AD, then broadly decrease from ~1400 AD until the present day (Fig. 7). Decadal variability is superimposed on the long-term decreasing trend (Fig. 7). Prominent, multi-decadal low- MBT'$_{5ME}$ value events are apparent from 1420–1444, 1500–1520, 1593–1627, 1762-1829, and 1908–1950 AD (Fig. 7). Multidecadal high- MBT'$_{5ME}$ events are observed from 1261–1283, 1317–1351, 1556–1583, 1632–1657, 1829–1846 and 1958–1987 AD(Fig. 7). The brGDGT concentrations in the core range from 1.2-21.1 µg gsed$^{-1}$, with a median of 8.2 µg gsed$^{-1}$.

### 4.5 BrGDGT isomer ratios

The UHPLC method we utilized for brGDGT analysis (Hopmans et al., 2016) allows for the separation of 5- and 6-methyl brGDGT isomers (DeJonge et al., 2013; DeJonge et al., 2014). Analysis of the relative abundances of these isomers has also been used to identify different production sources of brGDGTs (e.g. soils vs. water column; DeJonge et al., 2014; Weber et al., 2015). The summed IR (equation 3) for soils, lake water and sediments at Basin Pond are significantly different. The IR value for soils is very low and averages 0.03, while the water samples and sediments are higher and average 0.26 and 0.30, respectively (Fig. 6).

### 4.6 Algal biomarkers

Samples in the upper 17cm of the sediment core were analyzed for the following algal lipid biomarkers: (iso)loliolide, C$_{30}$ 1, 13 *n*-alkyl diol, dinosterol/stanol, and β-sitosterol/stanol.

Concentrations of (iso)loliolide ranged from 31-426 µg gsed$^{-1}$ (median=171). $C_{30}$ 1, 13 $n$-alkyl diol concentrations ranged from 8-1378 µg g sed$^{-1}$ (median=475). Dinosterol/stanol concentrations ranged from 12-5663 µg g sed$^{-1}$ (median= 1384). Concentrations of β-sitosterol/stanol concentrations ranged from 3-7955 µg g sed$^{-1}$(median=1465).

## 5 Discussion

### 5.1 Sources and seasonal production of Basin Pond brGDGTs

It is important to constrain brGDGT sources before interpreting lacustrine sedimentary records. Multiple lines of evidence suggest brGDGTs deposited in Basin Pond sediments are predominantly
produced within the water column, in agreement with prior studies (e.g. Tierney and Russell, 2009; Buckles et al., 2014; Loomis et al., 2014). First, we observe significant differences in the fractional abundances of brGDGTs between soil, SPM and lake sediments (Figs. 2, 5), suggesting *in situ* production occurs in both soil and lacustrine environments, but that soil-derived brGDGTs do not exert a large influence on the Basin Pond sedimentary record. Average distributions of brGDGTs
reveal that lake sediments and SPM are similar in group I and III content, while the soils differ substantially (Figs 2, 5). The relative amounts of 5- and 6-methyl brGDGTs also differs between soils and lake water and sediment (Figs 2, 5), and as seen in the average IR values (Fig. 6). MBT'$_{5ME}$ and CBT'$_{5ME}$ values for lake sediment and soils are distinct, while SPM samples are similar to lake sediment samples, consistent with *in situ* brGDGT production within Basin Pond
(Fig. 6). The degree of cyclization (mean CBT'$_{5ME}$ = 1.2) is significantly lower in lake sediments than in soil samples (mean CBT'$5_{ME}$= 1.5) (p value = 0.021 from two-tailed t-test), and brGDGTs are more methylated (p value = 0.003) in lake sediments (mean MBT'$_{5ME}$ = 0.38) than in soils (mean MBT'$_{5ME}$ = 0.7) (Fig. 6). This agrees with differences in brGDGT distributions recorded in other temperate (Tierney et al., 2012; Wang et al., 2012; Loomis et al., 2014) and tropical (Tierney
and Russell, 2009; Loomis et al., 2012; Buckles et al., 2014) lakes and catchment area soils, and suggests *in situ* production of relatively more cyclized and methylated brGDGTs within lakes. In agreement with previous studies, we also note higher brGDGT concentrations in lake sediments (median – 8.2 µg gsed$^{-1}$) in comparison to watershed soils (median= 2.2 µg gsed$^{-1}$) (e.g. Sinninghe Damsté et al., 2009; Tierney and Russell, 2009) pointing to *in situ* brGDGT production.

BrGDGT fluxes at all depths are generally low throughout the summer months (June–July). A large flux increase at depth (18–30 m) occurs during September, when the lake is strongly stratified. Thus any transfer of brGDGTs from lower depths to the upper water column likely would be minimal. This suggests annual seasonal production of brGDGTs in Basin Pond, with a

fall bloom occurring at intermediate (18–30 m) depths. Therefore, brGDGT temperatures recorded in the lake sediments likely reflect a seasonally biased (fall), rather than mean annual, temperature. We make the following observations based on these results. First, peak brGDGT flux is observed at 18-30 m water depth, suggesting that the organisms producing the most brGDGTs thrive in the mid to upper water column (Fig. 3). Secondly, peak brGDGT production occurs in September,

suggesting that the sedimentary record will be biased toward brGDGTs produced during this period (Fig. 3). Finally, for the four time-periods sampled, brGDGT distributions (as described by MBT'$_{5ME}$) correlate with temperature (Fig. S1). Interestingly, at depth the water temperature shows little to no seasonal cycle, remaining at approximately 4 °C for the entire year (Frost, 2005). Therefore, if maximum brGDGT production is indeed occurring here, it is possible that another

parameter, which covaries with temperature on a seasonal scale (i.e. light duration, water chemistry, nutrient availability), may drive, or contribute to, the distribution of brGDGTs produced at depth at Basin Pond. However, the sediment trap at this depth represents an integrated signal of SPM produced in the water column, which could also be driving the temperature correlation at depth.

Although few studies are available for comparison, Loomis et al. (2014) studied brGDGTs in another temperate lake in the NE US (Lower King Pond, Vermont). brGDGT production in Lower King Pond peaked during fall and spring and was linked to seasonal full water column mixing events (Loomis et al., 2014), which do not occur at Basin Pond (Frost, 2005). Moreover, Basin Pond is ten times larger by area and four times deeper than Lower King Pond. Similar to Lower

King Pond, brGDGT production at Basin Pond seems to be seasonal, and calibration of brGDGTs against seasonal (in this case, fall) temperatures is a necessary area of future work to accurately reconstruct past absolute temperature change for this location. If the behaviour of brGDGTs in Lower King Pond and Basin Pond is representative of all temperate lakes, then calibration to fall or spring temperature may be the most appropriate choice for these settings.

## 5.2 Calibration of the 900 year brGDGT record to temperature

Numerous studies have provided strong evidence for *in situ* brGDGT production in lakes and have shown that application of the global soils calibration to lacustrine sediments often yields temperatures that are unrealistically cold (e.g. Tierney and Russell, 2009; Bechtel et al., 2010;

Blaga et al., 2010; Tierney et al., 2010a,b; Tyler et al., 2010; Pearson et al., 2011). Therefore, many lacustrine brGDGT calibrations have been developed (Tierney et al., 2010; Zink et al., 2010; Pearson et al., 2011; Sun et al., 2011; Loomis et al., 2012, Foster et al., 2016). However, many of these are based on relatively few samples or are geographically restricted (e.g. Tierney et al., 2010; Zink et al., 2010; Foster et al., 2016). Furthermore, at present, all available lacustrine brGDGT

calibrations except for two (Dang et al., 2018; Russell et al., 2018) were developed using older HPLC methods that did not fully separate brGDGT isomers. As we measured our brGDGTs following the newer method of Hopmans et al. (2016), we investigated Basin Pond temperature reconstructions using only those calibrations based on the same technique. The Dang et al. (2018) calibration is based on alkaline Chinese lakes and reconstructs temperatures ranging from 4-9 ℃,

while the Russell et al. (2018) calibration is based on African lakes and yields temperatures ranging from 10-14 ℃ (Fig. 7). The African lakes calibration from Russell et al. (2018) is based on MBT'$_{5ME}$ while the Chinese lakes calibration of Dang et al., (2018) is based on fractional abundances of brGDGTs; therefore these two calibrations yield somewhat different trends with the Dang et al. (2018) calibration showing muted variability and some discrepancies from the other

proxy records (i.e. during the last 50 years) (Figure 7).

Importantly, caution must be taken when interpreting the Basin Pond reconstructed temperatures using either of these calibrations because application of an African or Chinese calibration to lakes in the NE US is questionable as these regions are climatically different and their lakes differ in terms of stratification and mixing regimes. Furthermore, brGDGTs from Basin Pond are

characterized by distinct brGDGT distributions from both the African (Russell et al., 2018) and alkaline Chinese lake sediments (Dang et al., 2018) (Figure 8). This suggests that application of either of these calibrations to Basin Pond sediments may not be appropriate. Local temperature data are available for Basin Pond over the period our measurements were made, but the SPM dataset presented here is not large enough to develop a robust local MBT'$_{5ME}$ to temperature

calibration (see Supplement). We thus present our results in the following discussion and figures

simply in terms of the MBT'$_{5ME}$ index, which provides a relative temperature indicator, where higher values reflect relatively higher temperatures and vice versa (De Jonge et al., 2014b).

Our interpretation of the 900-year MBT'$_{5ME}$ record is as follows. Based on the SPM samples, we argue that the downcore brGDGT reconstruction is likely weighted toward September temperature change in the NE US. We note that brGDGTs are present at all depths measured but peak at 18-30 m depth, indicating that the compounds reaching the lake floor represent an integrated signal from the entire water column. Although brGDGT concentrations vary down core, they are not correlated with reconstructed MBT'$_{5ME}$ values (p=0.25), indicating that brGDGT production and MBT'$_{5ME}$ variability are largely decoupled. We observe an overall stepped cooling trend recorded by generally decreasing MBT'$_{5ME}$ values over the past 900 years (Fig. 9). Using the calibration of Russell et al. (2018), this overall cooling is on the order of 3.0 $^{o}$ C; however, for the reasons discussed earlier, we advise that caution must be taken when interpreting absolute temperature changes from applying this calibration to Basin Pond sediments.

**5.3 Comparison to Regional Hydroclimate Records in the NE US**

Regional hydroclimate in the NE US has been reconstructed at several sites on similar timescales as the Basin Pond record. The MBT'$_{5ME}$ record indicates an overall cooling from ~1300 AD to ~1900 AD (Fig. 9), which is also observed in pollen-derived temperature reconstructions from Basin Pond (Gajewski, 1988). Both records also indicate two major cooling steps, although the exact timing of these differs between records, which may be attributable to age model differences (Fig. 9). Apparent differences between the Basin Pond records are likely also associated with sampling resolution;  the pollen record has varying and generally much lower sample resolution in comparison to our MBT'$_{5ME}$ record. Furthermore, some of the differences in MBT'$_{5ME}$ and pollen reconstructions may be caused by differences in proxy seasonality, with pollen representing a summer signal (Gajewski, 1988) and MBT'$_{5ME}$ likely representing a fall signal.

The general long-term cooling trend from Basin Pond is also observed in a hydrogen isotope-based temperature reconstruction from Little Pond, Massachusetts (Gao et al., 2017). Both records show higher temperatures between 1300-1400 AD (Fig. 9). Bog records provide additional, high-resolution reconstructions of hydrological conditions in the NE US over this time period via analysis of testate ameoba (a proxy for water table depth) and the *Sphagnum*/Vascular Ratio (SVR) (Nichols and Huang, 2012; Clifford and Booth, 2013). The testate ameoba records show that the

last 400 years (i.e., 1600–2000 AD) have been generally wetter than the preceding 400 years (1200-1600 AD). However, unlike the temperature reconstructions, these records do not show a long-term linear trend (Fig. 9).

These cooling and wetting trends are surprising given the record of fire history at Basin Pond
(Miller et al., 2017), which shows five periods of increased charcoal deposition since 1100 AD (Fig. 9). It is important to note that wildfire activity is a complex phenomena, with multiple factors affecting fire occurrence apart from climate variability (Marlon et al., 2017). However, our data suggest that fire activity in the NE US may be influenced more by shorter-term (multi-decadal) variations in climate, particularly seasonal cooling superimposed on dry conditions, as opposed to
longer-term, multi-centennial climate trends. Surprisingly, the first 200 years of the record (1100–1300 AD) are dominated by warm and dry conditions, but no fire events were recognized during this period. Three fire events (Fig. 9g), between ~1300 and ~1700 AD, are associated with regionally dry conditions (Fig. 9d-f). Although average Basin Pond MBT'$_{5ME}$ values are higher on a multi-centennial time-scale during this interval, the fire events themselves occur synchronously
with multi-decadal cold periods (Fig. 9a-c). Furthermore, two recent fire events occurred during the historical period, which is reconstructed as relatively cool and wet (Fig. 9). Therefore, it appears that at Basin Pond, temperature did not exert a major influence over fire occurrence.

**5.4 Comparison with Northern Hemisphere records**

A compilation of Northern Hemisphere temperature records for the last 2000 years reveals
sustained warmth from 830–1100 AD, just prior to the beginning of our reconstructions (PAGES2k, 2013). Northern Hemisphere climate then entered a cooler phase, though the timing of this transition varied regionally between 1200 and 1500 AD (PAGES2k, 2013). North American pollen data show elevated, though decreasing, temperatures through 1500 AD (Gajewski, 1988; PAGES2k, 2013) (Fig. 10). From 1100-1400 AD, Basin Pond MBT'$_{5ME}$ values are high in contrast
with European and Arctic temperature reconstructions. From 1500-1900 AD, MBT'$_{5ME}$ values are lower, in better agreement with other Northern Hemisphere reconstructions. Moreover, the decadal to centennial scale variability observed in MBT and other records during this time may be linked to variability in Atlantic Multidecadal Oscillation (AMO) and North Atlantic Oscillation (NAO) indices (Figure 8). We note that the brGDGT MBT'$_{5ME}$ values are better correlated with regional
tree-ring records and compilations of European and Arctic temperatures, which all show warm

anomalies, followed by cooling, earlier this century. Thus, the multi-centennial structure of the brGDGT record from Basin Pond is supported by other local, regional, and global records (PAGES2k, 2013). On a multi-decadal scale, there is variability potentially associated with volcanic events recognized as having a global impact. Five intervals during the last millennium were defined as 'volcanic-solar downturns': 1251–1310, 1431–1520, 1581–1610, 1641–1700, and 1791–1820 AD (PAGES2k, 2013). All but the most recent (1908–1950 AD) of the cool events are present in the Basin Pond MBT'$_{5ME}$ record during these periods (or within the age model uncertainty) (Fig. 10 highlighted in blue).

There is some similarity between the Basin Pond reconstruction and other Northern Hemisphere reconstructions (PAGES2k, 2013) (Fig. 10). The brGDGT record is also peppered with warm (high MBT'$_{5ME}$) anomalies; many of these seem to be coherent with tree-ring based reconstructions of North American climate (i.e. 1300, 1550, 1830 AD) and are sometimes associated with negative phases of the NAO (Fig. 10). The sensitivity of the Basin Pond sediment record to regional scale climatic variations is highlighted by time series analysis. Multispectral taper method analysis reveals a persistent cycle in the brGDGT-based temperature reconstruction with a period of 57–63 years (Fig. 10). The Northern Hemisphere tree-ring compilation also shows a cyclicity with a period of 60 years (Fig. 10). However, the fact that the two datasets are not significantly correlated indicates the variability at 60-year periods is not exactly in-phase over the 900 year period covered by the two records. Cross-correlation analysis indicates that the correlation between the two datasets is strongest when the tree-ring reconstruction is lagged by 42 years relative to the Basin Pond temperature record (r=0.33, p=0.04). Significant spectral peaks with a similar period exist in the annually-resolved records from Europe (period = 65 yr), Asia (period = 58 yr) and the Arctic (period = 58 yr) (PAGES2k, 2013). However, the same analyses applied to South American, Australasian, and Antarctic reconstructions do not show spectral peaks at this period (PAGES2k, 2013). Thus, it appears that the Basin Pond brGDGT record captures variability that is representative of, but not necessarily in-phase with, the Northern Hemisphere at large. One possible mechanism to explain this is the North Atlantic Oscillation (NAO), which exhibits a quasi-periodic oscillation of ~60 yr (Sun et al., 2015). While the NAO has some regionally coherent climatic effects, the signature of positive and negative NAO modes is spatially heterogeneous and complex; this could explain the phase offset in the ~60 yr band between the Basin Pond record and the other Northern Hemisphere reconstructions.

Another possible driver of the MBT'$_{5ME}$ changes we see is the AMO, which is based on sea surface temperature anomalies in the North Atlantic and shows variability in quasi-periodic 60–80 yr cycles (Trenberth et al., 2017). An AMO reconstruction spanning the last 400 years shows some similarities to the MBT'$_{5ME}$ reconstruction from Basin Pond. Although the records do not show a

strong cross correlation (r=0.08, p=0.53), they feature apparently synchronous cool and warm periods (i.e. 1550 - 1650 AD and 1780 – 1830 AD) (Fig. 10). This suggests that climate at Basin Pond is coupled to Atlantic sea surface temperatures on multi-decadal timescales. Thus, the record presented here may prove useful in the future for reconstructing changes in the AMO earlier in the paleorecord.

**5.5 20th century Meteorological station and Maine temperature data**

Daily temperature averages from meteorological stations in the state of Maine were accessed and obtained through the National Climatic Data Center from 1895 – present day (NOAA, 2014) (Fig. 11). Average temperatures in Maine have warmed by ~1.5ºC since 1895 AD (Fig. 11). The temperature increase is dominated by changes from 1895-1945 AD and 1985-present; for the forty

intervening years mean temperatures were more stable, with a slight cooling observed during fall (Fig. 11). Interannual variability of +/- 1ºC is observed throughout the record, with the most pronounced variability during the winter (NOAA, 2014).

**5.6 20th Century brGDGT Reconstructions and Algal Community Shifts**

Interestingly, variations in MBT'$_{5ME}$ values for the last 100 years do not agree with instrumental

observations. The brGDGT-based reconstruction shows stable values from 1900-1950 AD, followed by an abrupt increase (warming) in MBT'$_{5ME}$ of 0.1 until approximately 1975 AD (three data points), and a subsequent continual decrease since then (five data points) (Fig. 11). In contrast, instrumental records indicate a slight cooling, or at least a stabilization of warming, starting at the same time (1960s-70s) when the MBT'$_{5ME}$ values are increasing (Fig. 11). We note decreasing

MBT'$_{5ME}$ values in the upper 3.5 cm (Fig. 11). Low MBT'$_{5ME}$ values in surface and core top sediments have been noted in other studies as well (e.g. Sinninghe Damsté et al., 2012; Tierney et al., 2012), indicating that this feature occurs in different regions and environments, and may be driven by mechanisms associated with brGDGT production or preservation. Tierney et al. (2012) note a similar pattern in the brGDGT distributions of Salt Pond (RI) surface sediments that we

observe at Basin Pond where the shallow surface sediments are characterized by more methylated brGDGTs. These authors suggest that more methylated brGDGTs present in shallow lake sediments do not survive diagenesis and they also note that deeper sediments yielded reasonable brGDGT reconstructed temperatures (Tierney et al., 2012). This hypothesis requires further testing

and additionally, other influences such as changes in brGDGT producer, post-depositional mobility and/or overprinting of the brGDGT signal, biotic and abiotic compound diagenesis, and anthropogenic impacts to lake ecosystems should be examined as well. Despite the uncertainties about the MBT'$_{5ME}$ record during the last 100 years, we believe that the Basin Pond brGDGT record is useful for describing regional climate evolution over the last millennium in the NE US.

It is possible that land-use change and other anthropogenic impacts have affected the brGDGT record over the last 100 years. However, known land use change in the Basin Pond catchment is minimal over the past century (Gajewski, 1988). A complicating factor is the addition of the piscicide rotenone to the lake in 1955 to remove fish species in competition with trout (USGS, 1996). While the estimated lifetime of rotenone in the water column is short (days to weeks), it

has been shown to have lasting long-term (years) effects on zooplankton communities and lake productivity (Kiser, 1963; Andersen, 1970; Sanni and Waervagen, 1990).

In lacustrine environments, some classes of lipid biomarkers, specifically sterols and stanols, can give valuable insight into variability of lake productivity of certain types of algae throughout time. Many sterols (and their saturated counterparts, stanols) are indicative of certain groups of source

organisms, in particular, specific phytoplankton groups (e.g. Volkman et al., 1998; Volkman, 2003). For example, dinosterol and dinostanol are found in dinoflagellates and are not produced in higher plants, and are therefore used as a biomarker for dinoflagellate species (Volkman et al., 1998). The compounds isololiolide and loliolide are known to be anoxic degradation products of diatom pigments (Klok et al., 1984; Repeta, 1989) while long-chain alkyl diols are produced by

eustigmatophyte (yellow-green) algae (Volkman et al., 1998). At Basin Pond, several algal biomarker concentrations, including isololiolide/loliolide, dinosterol/stanol, and $C_{30}$ 1,13 *n*-alkyl diol, decrease following the rotenone treatment in 1955 AD while β-sitosterol (a biomarker of higher plants) increases (Fig. 11) suggesting a shift in the overall algal community structure. Additionally, after 1955 contributions of the different algal biomarkers are remarkably stable in

comparison to earlier times (Fig. 11). Due to the widespread shift in algal community, we posit that bacterial communities and therefore brGDGT production may also have been impacted.

However, brGDGT concentrations do not clearly respond to the rotenone treatment, and additional knowledge of brGDGT producers would be required to further investigate this idea.

**6 Conclusions**

We find evidence for seasonally-biased, *in situ* production of branched gylcerol dialkyl glycerol tetraethers (brGDGTs) in a lake in central Maine, NE US. BrGDGTs are mostly produced in September at Basin Pond, and their downward fluxes in the water column peak at 30 m water depth. A downcore brGDGT-based reconstruction reveals both gradual and transient climate changes over the last 900 years and records cooling and warming events correlated with other Northern Hemisphere records and the NAO and AMO indices. This suggests inland Maine climate is sensitive to hemispheric climate forcing as well as changes in regional atmospheric pressure patterns and North Atlantic sea surface temperatures. Our new MBT'$_{5ME}$ temperature reconstruction, supported by a pollen record from the same site, reveals a prominent cooling trend from 1100–1900 AD in this area. Comparison with regional hydroclimate records suggests that despite increasingly cool and wet conditions persisting at Basin Pond over the last 900 years, fire activity has increased. Although recent fire activity is likely anthropogenically triggered (i.e. via land-use change), our results imply an independent relationship between climate and NE US fire occurrence over the study interval. Thus, the paleotemperature reconstruction presented here alongside site-specific knowledge from Basin Pond informs our understanding of climatic variability in NE US beyond the era of human influence.

**Data Availability**

BrGDGT data, including fractional abundances of 5- and 6-methyl isomers, BIT Index values, MBT'$_{5Me}$ values, CBT'$_{5Me}$ values, 5-methyl isomer ratio (IR), total brGDGT concentrations, and temperature calibrations (Dang et al., 2018; Russell et al., 2018) from Basin Pond watershed soils, SPM, and sediment samples are available at the National Oceanic and Atmospheric Administration National Centers for Environmental Information (NOAA NCEI) Paleoclimate Database. Concentrations of isololiolide/loliolide, C30 1,13 Diol, sitosterol/sitostanol, and dinosterol/dinostanol from the Basin Pond sediment core are also provided where measured. To

access these data, please visit: https://www.ncdc.noaa.gov/data-access/paleoclimatology-data/datasets.

## Author Contributions

5 DRM, MHH, and BAK designed the sediment traps, carried out field work, processed samples through all stages of laboratory prep and analysis, and prepared the manuscript for publication. ISC and RSB provided advice throughout the experiment and writing process, aided with field work, contributed to data interpretation, and covered costs of sample analysis. Manuscript revisions were made through contributions from all co-authors.

## Competing Interests

The authors declare that they have no conflict of interest.

15 **Acknowledgements**

The authors thank John Sweeney for his assistance in sediment trap construction and design. Jeff Salacup is acknowledged for technical laboratory assistance. We thank the members of the UMass Biogeochemistry Lab for helpful feedback and discussion. We are grateful to Lucas Groat, Christopher Mode, and Paige Miller-Hughes for their assistance in sediment trap deployment and 20 collection. Dr. Mike Retelle, Dan Frost, Julie Savage, and the Bates College Geology Department are recognized for sediment coring assistance. We are indebted to the Basin-David-Tilton Ponds Association for their cooperation and support. Funding for this project was provided by grant G12AC00001 from the United States Geological Survey and the 2014 and 2015 Joe Hartshorn Memorial Award through the UMass Amherst Department of Geosciences. We thank J. Hou and 25 an anonymous reviewer for their comments. We also acknowledge Liping Zhou and Denis-Didier Rosseau for editing this manuscript. Results are solely the responsibility of the authors and do not necessarily represent the views of the Northeast Climate Adaptation Science Center or the USGS.

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

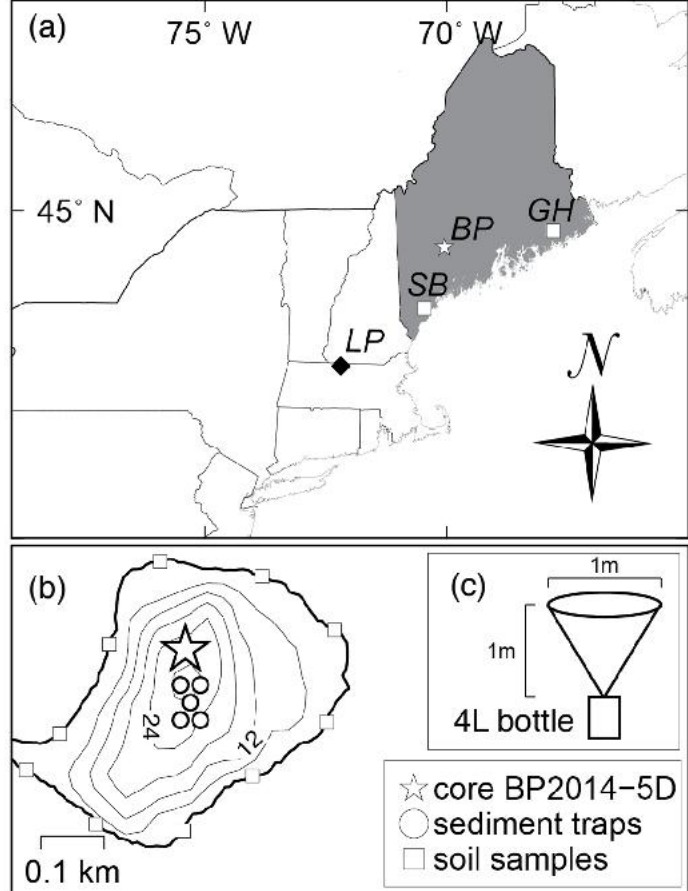

**Figure 1. Map of Basin Pond. (a) The location of Basin Pond (BP) (white star) in Maine, USA. Locations of three other sites are labelled: Little Pond (LP; Gao et al. 2017), Great Heath Lake (GH; Nichols and Huang, 2012; Clifford and Booth, 2013), and Saco Bog (SB; Clifford and Booth, 2013). (b) Bathymetric profile (6 m contours) of Basin Pond with position of floating sediment traps (circles), surface soil samples (squares), and core BD-2014-5D used for the downcore temperature reconstruction in this study (star). The pond has an area of approximately 0.14 km². (c) Schematic of sediment traps utilized in this study.**

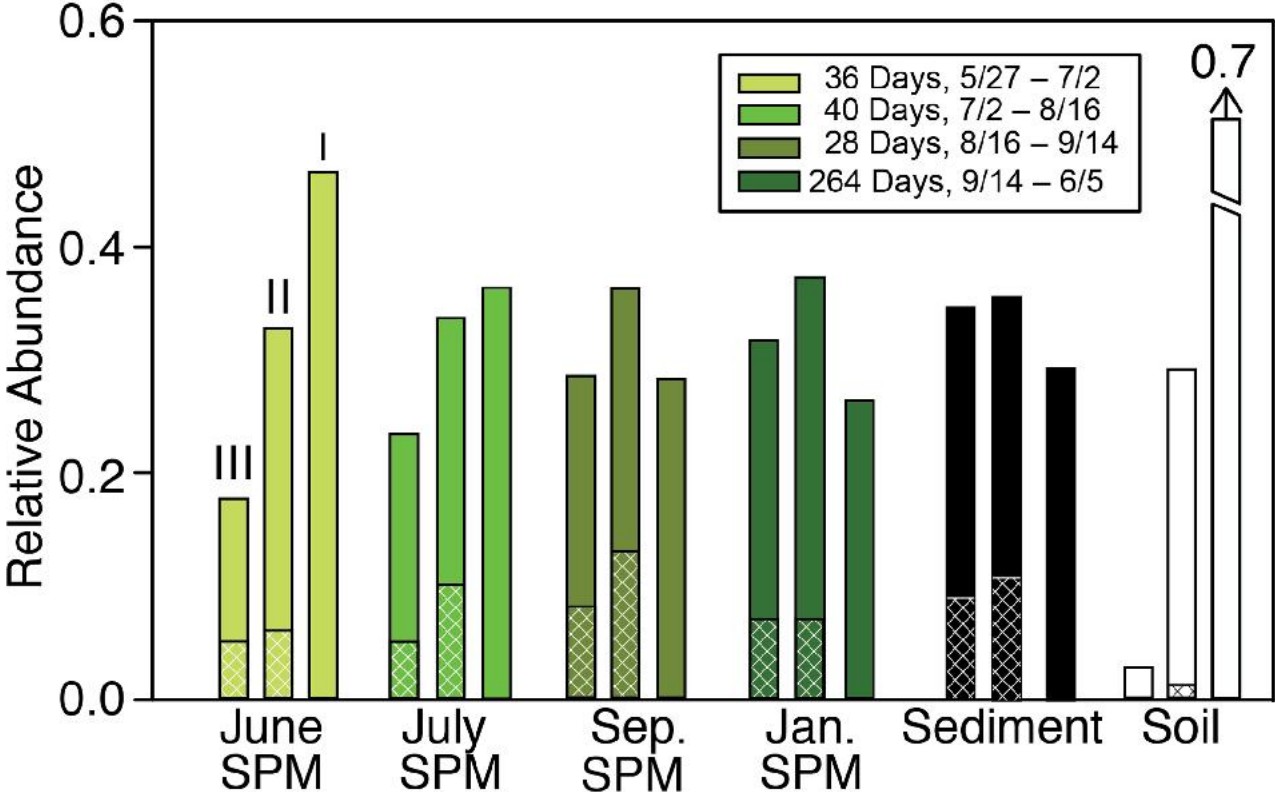

**Figure 2. Temporal variation of the relative abundance of group I, II and III brGDGTs in SPM (green shaded bars), sediment (black bars), and catchment soil samples (white bars). As in the plot of June SPM, the brGDGT groups III, II, and I, are displayed from left to right for each collection period and sediment and soil samples. Sediment and soil samples were collected in Spring of 2014. Green shaded bars for SPM samples reflect averages for each date samples were collected, measured in July 2014 (lightest green), August 2014, September 2014 and June 2015 (darkest green). For each category, brGDGT groups III, II, and I are shown in that order (left to right). Lines in each bar represent the relative abundance of 5- and 6- methyl brGDGTs, with cross hatching representing 6- methyl abundances.**

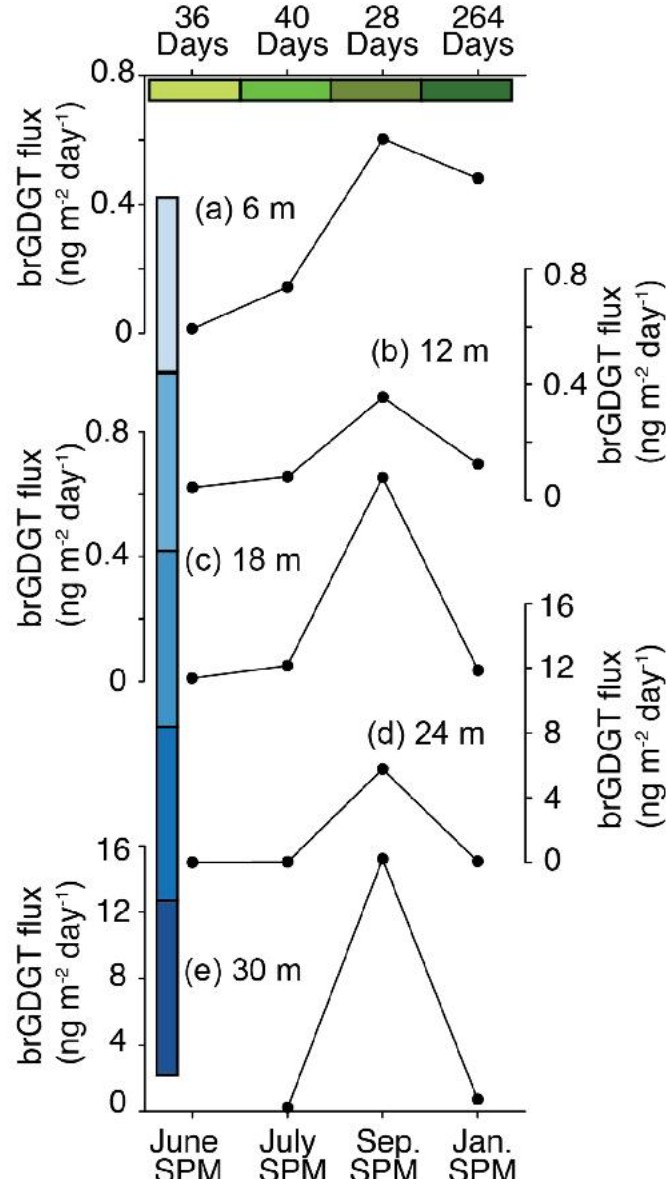

**Figure 3. Time series of brGDGT fluxes for each of the sediment traps in Basin Pond. brGDGT fluxes at 6m (a), 12m (b), 18m (c), 24m (d), and 30m (e) are shown. There is no data for trap (e) in July 2014. Note the change of scale for (d) and (e), indicating fluxes an order of magnitude higher for the lowermost traps. Green bars correspond to the time periods in Figure 2. Blue bars correspond to the depth ranges in Figure 5.**

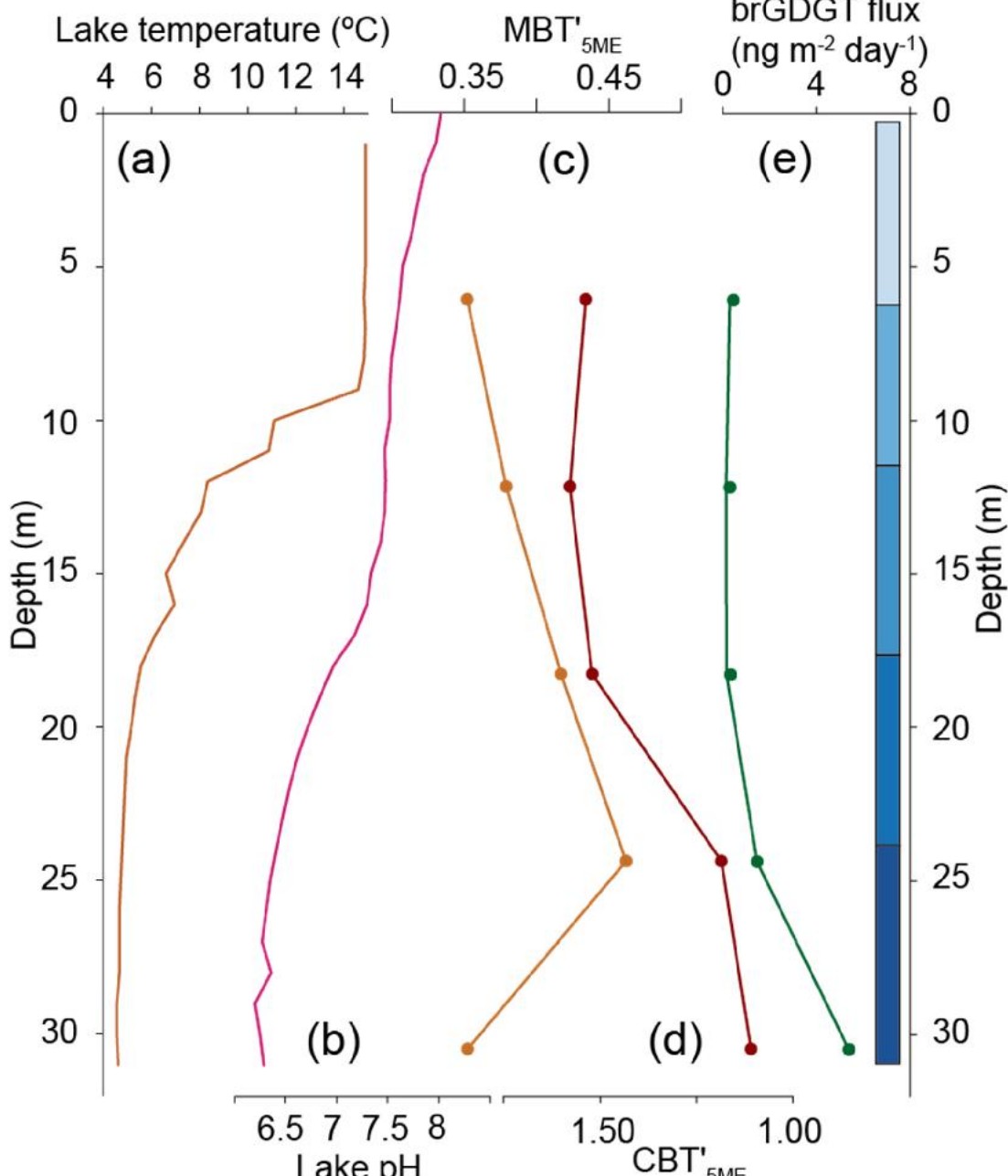

**Figure 4. Hydrolab-measured temperature and pH profiles for Basin Pond compared with flux weighted average brGDGT-based reconstructions. (a) Fall lake temperature profile, showing the mixed layer extending to ~9 m water depth, followed by the thermocline (9-15**
5   **m) and a cold deep layer (15-32 m). (b) Fall pH profile. the pH ranges from ~7.5 at the surface to ~6.2 at depth. (c) Flux weighted average MBT values measured at sediment traps. (d) Flux weighted average CBT values measured at sediment traps. (e) brGDGT fluxes measured at sediment traps.**

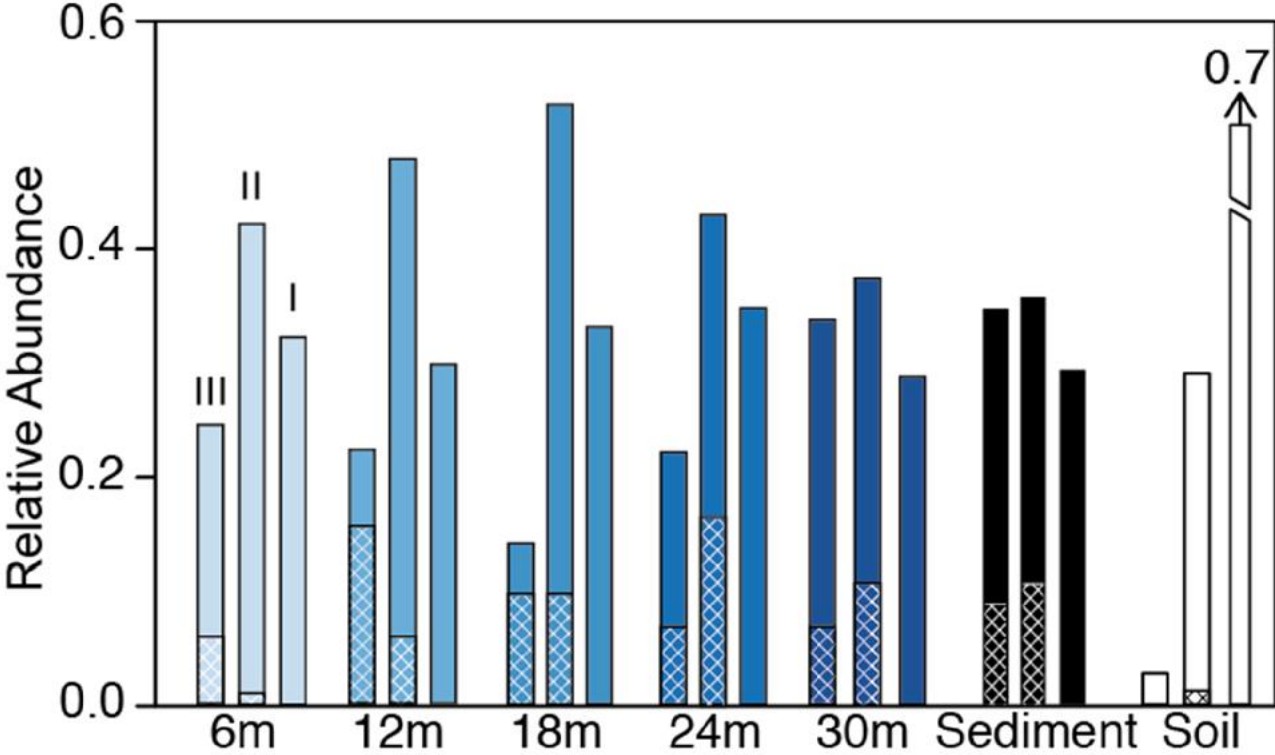

**Figure 5. Spatial variation in the water column of the relative abundance of groups I, II and III brGDGTs in SPM as a function of water depth. As in the plot of June SPM, the brGDGT groups III, II, and I, are displayed from left to right for each collection period and sediment and soil samples. For each group, the relative abundance at depths of 6 m (lightest blue), 12 m, 18 m, 24 m, and 30 m (darkest blue) is plotted next to the average surface sediment (black) and catchment soil (white). For each category, brGDGT groups III, II, and I are shown in that order (left to right). Lines in each bar represent the relative abundance of 5- and 6-methyl brGDGTs, with cross hatching representing 6- methyl abundances.**

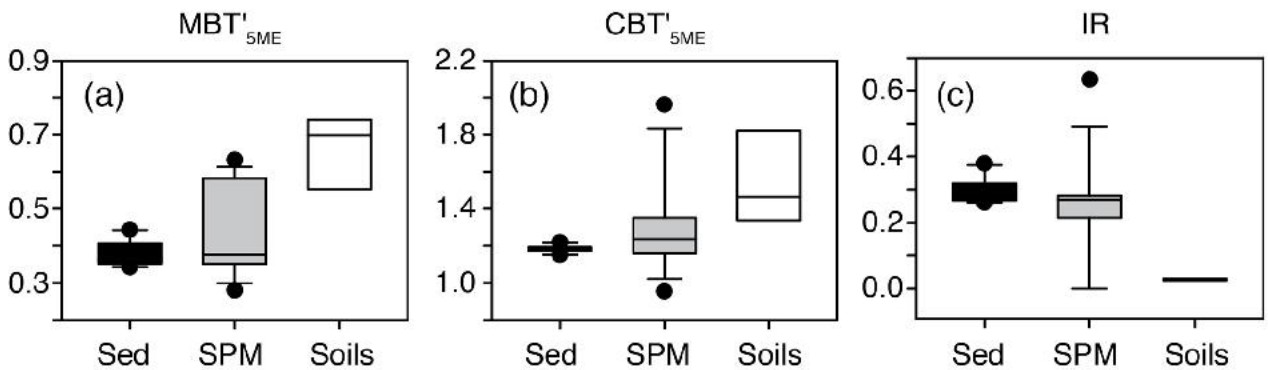

5 **Figure 6. BrGDGT-based proxies measured on surface sediments (black), SPM (gray), and catchment soils (white). (a) Cyclization of Branched Tetraethers (CBT), (b) Methylation of branched tetraethers (MBT'$_{5ME}$), and (c) the Isomer Ratio (IR).**

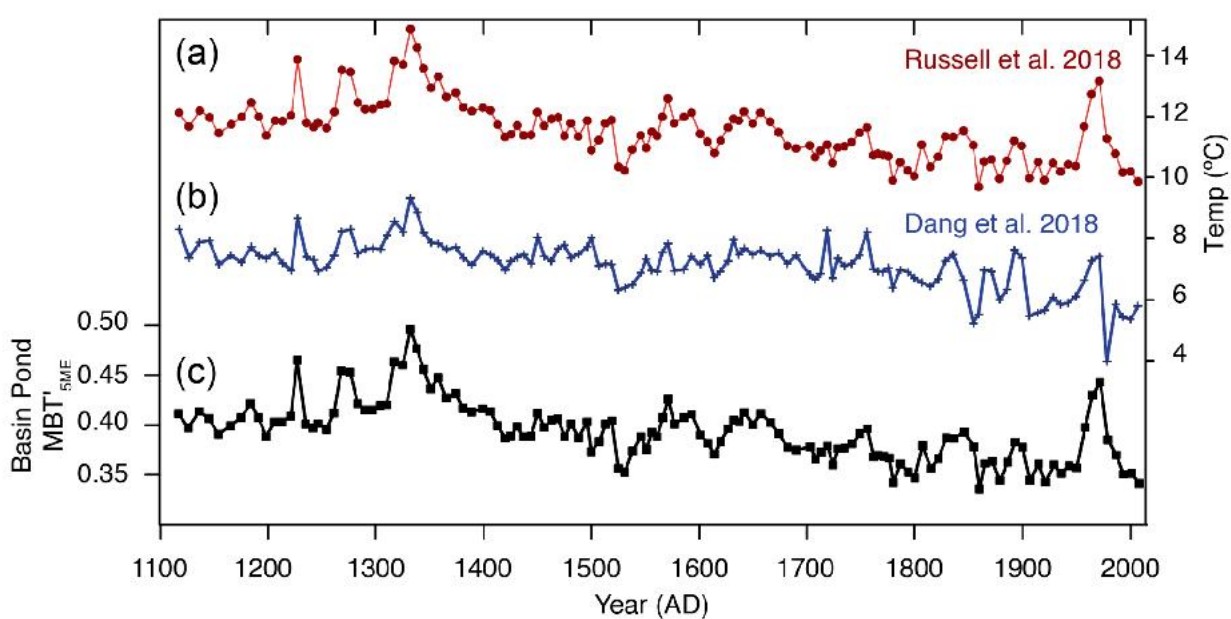

**Figure 7. Comparison of Basin Pond MBT'5ME with newly published temperature calibrations. (a) Core BP2014-5D plotted using the African Lakes calibration (Russell et al., 2018), and the (b) Chinese lakes calibration (Dang et al., 2018). (c) Basin Pond MBT'5ME values.**

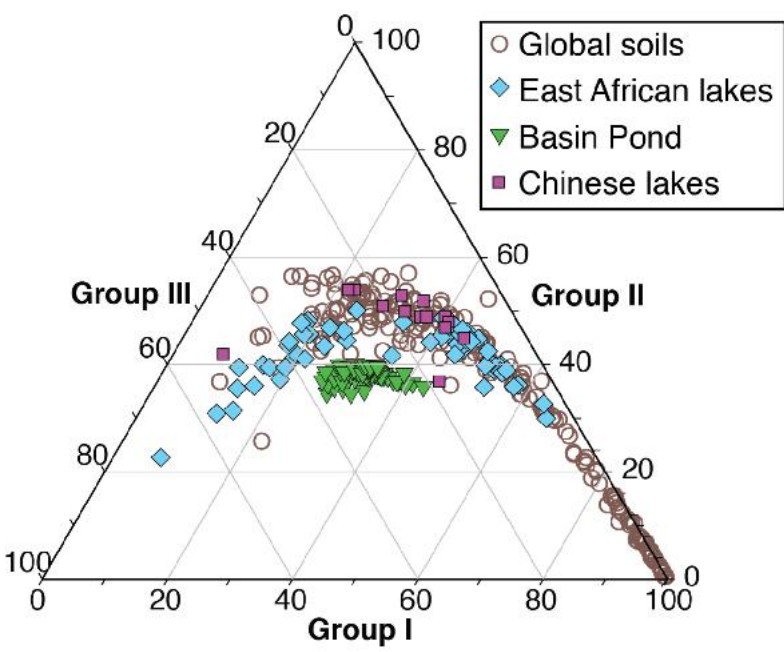

**Figure 8. Ternary diagram of brGDGT distributions of lake sediments (Dang et al., 2018; Russell et al., 2018) and global soils (Peterse et al., 2012) and Basin Pond sediments.**

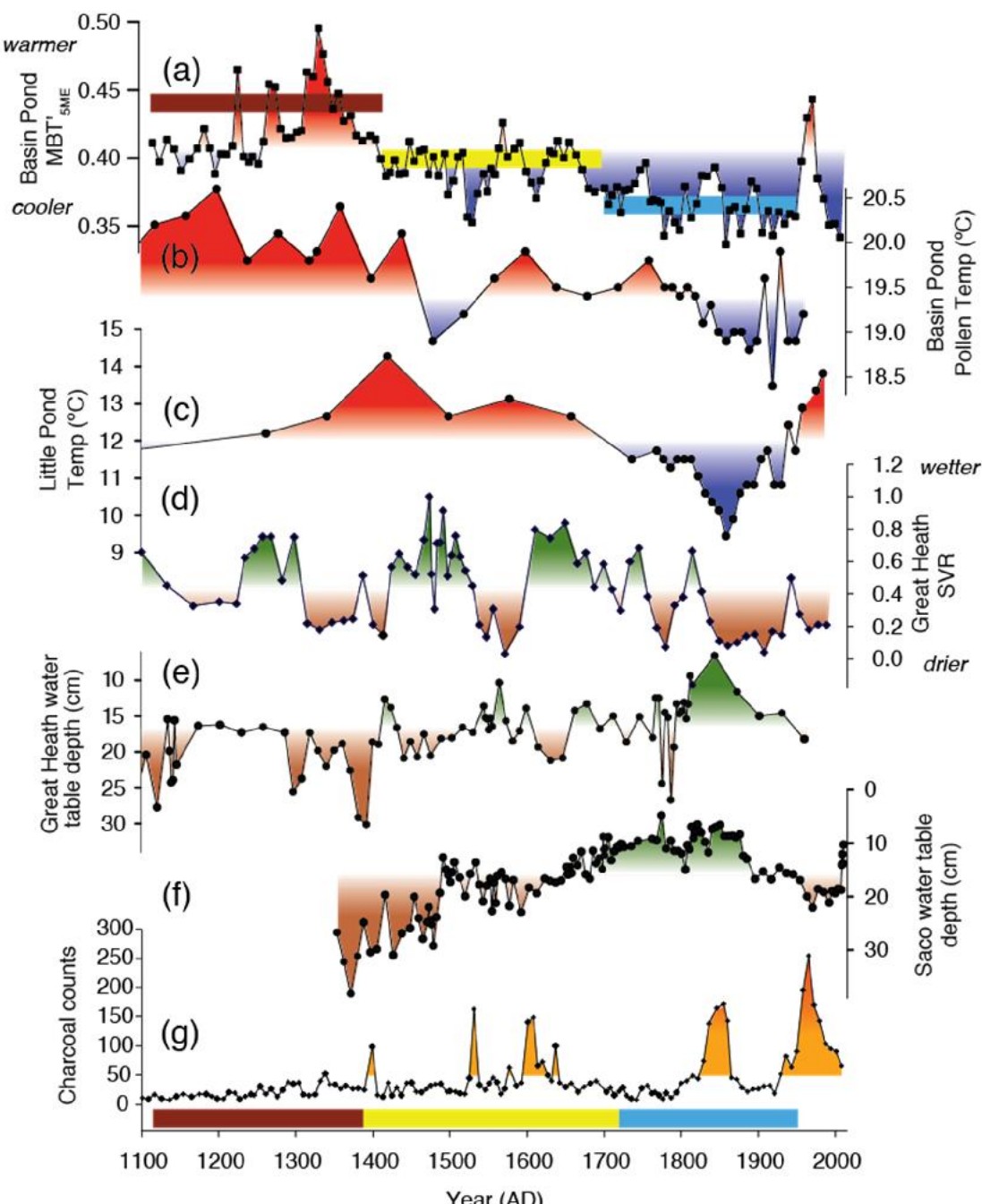

**Figure 9. The Basin Pond MBT's_{5ME} record compared with other paleoclimate records from the NE US. (a) MBT's_{5ME} (this study). Colored bars indicate the three main periods discussed in the text. (b) Pollen-based reconstruction of temperature at Basin Pond (Gajewski, 1988).**
5 **(c) Deuterium isotope (δD)-based temperature reconstruction at Little Pond (Gao et al., 2017). (d) Great Heath aridity reconstruction based on the *Sphagnum*/Vascular Ratio (SVR) (Nichols and Huang, 2012). (e) Water table reconstruction from Great Heath (Clifford and Booth, 2013). (f) Water table reconstruction from Saco Bog (Clifford and Booth, 2013). (g) Charcoal counts from Basin Pond (Miller et al., 2017).**

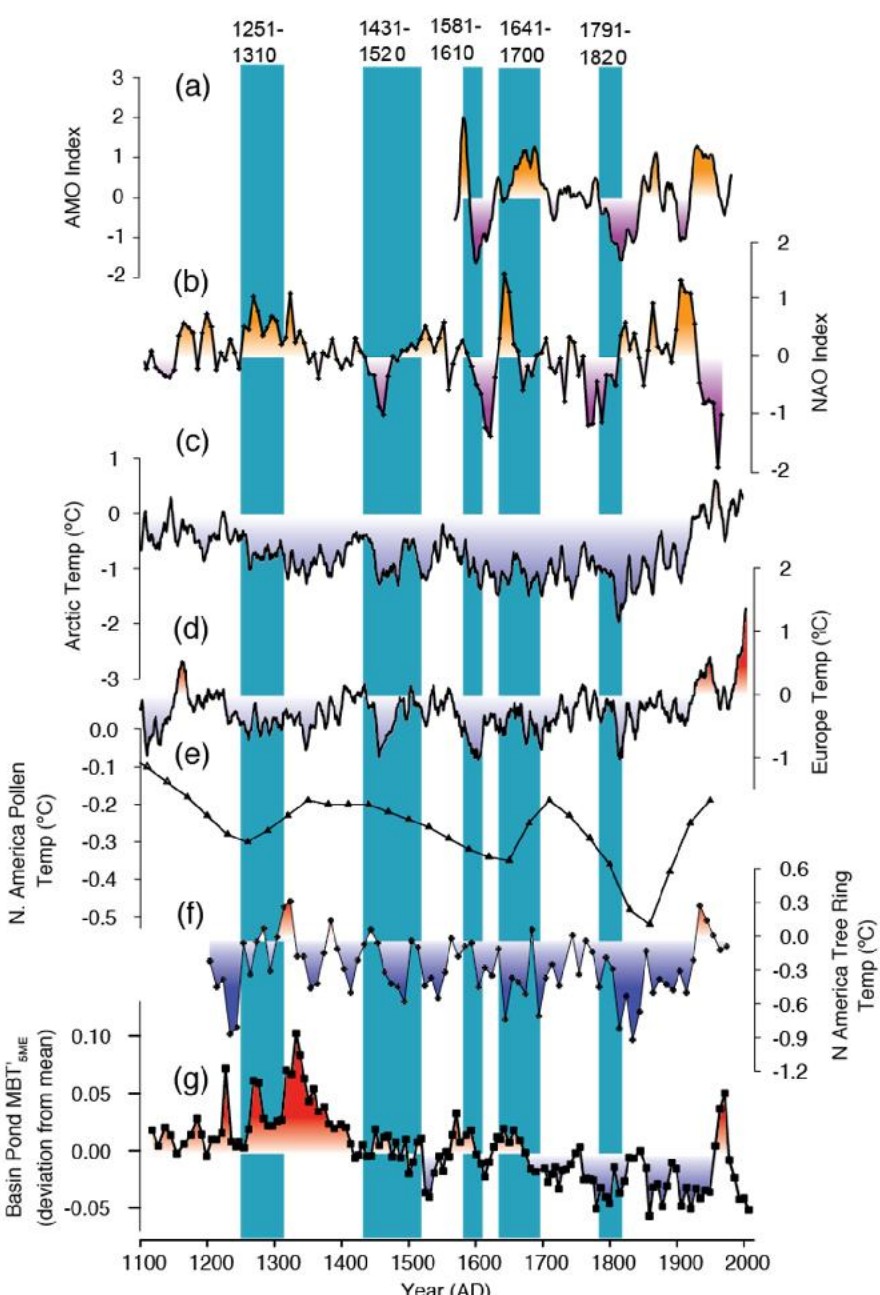

**Figure 10. The Basin Pond MBT'₅ME record compared with regional and global records of temperature change. (a) Tree-ring based reconstruction of the AMO Index (Gray et al., 2004). (b) NAO Index reconstruction (Sun et al., 2015). (c–f) Regional temperature stacks based on composite proxy reconstructions for the Arctic (c), Europe (d), and North America (pollen, (e); tree rings, (f). The records have been standardized to have the same mean (0) and standard deviation (1) from 1190–1970 AD (PAGES2k 2013). (g) MBT'₅ME (this study).**

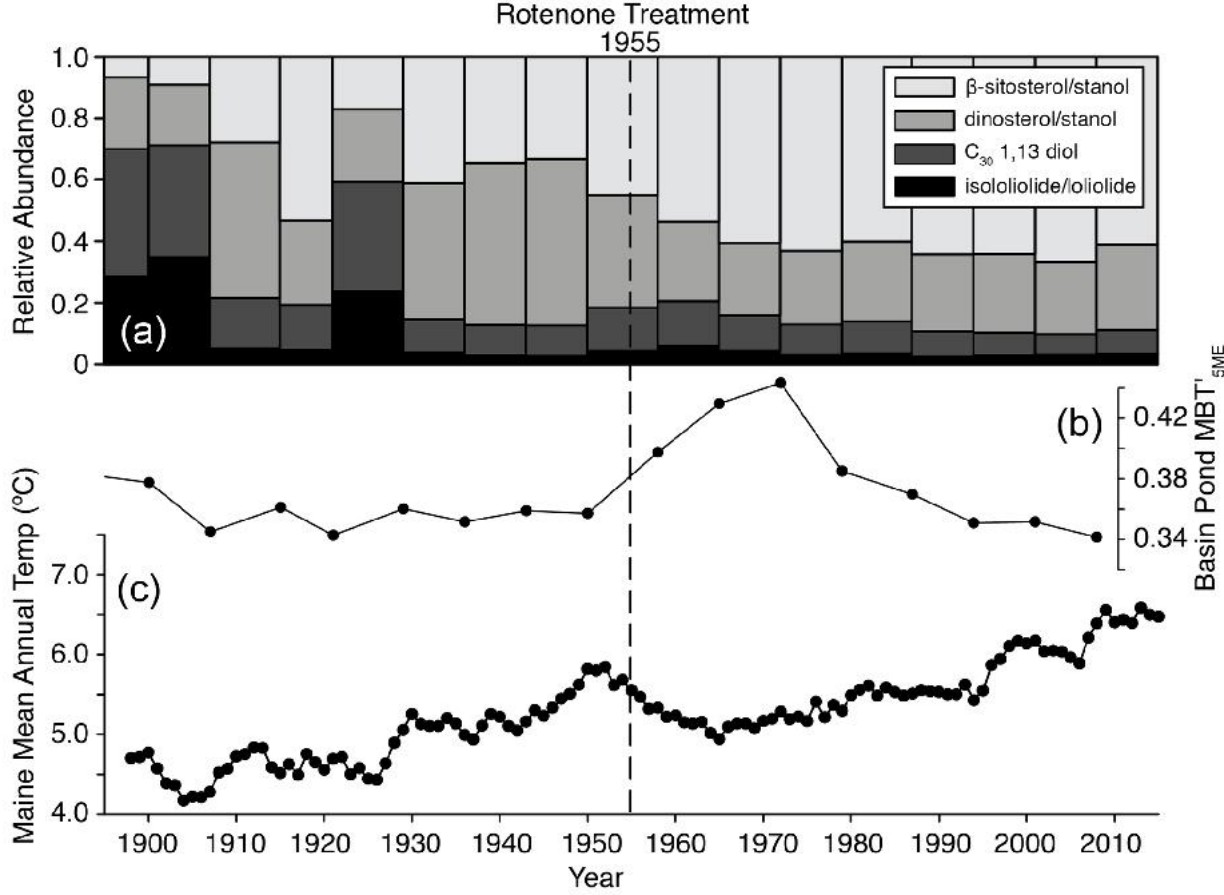

**Figure 11. Comparison of historical temperature records for the state of Maine, MBT'$_{5ME}$ reconstruction, and algal lipid biomarkers in Basin Pond. (a) Relative abundance of four major algal lipids. (b) MBT'$_{5ME}$ record. (c) Maine state-wide average temperature (NOAA, 2014). The black line indicates the rotenone treatment of the lake in 1955.**