# Peer review of "A 900-year New England temperature reconstruction from *in situ* seasonally produced branched glycerol dialkyl glycerol tetraethers (brGDGTs)"

_Climate of the Past, 2018_

## Referee Comment (RC1) · J. Hou (Referee) · 27 Apr 2018

Miller et al. presented a well-designed experiment to investigate the seasonality of brGDGT proxies in this paper, which is helpful to understand the mechanism of the potential temperature proxy. The authors reconstructed temperature variation in the past 900 years and suggested they could differentiate anthropogenic and natural changes. I think it is a good try to understand the seasonality of brGDGT proxies, which is worthy to be published. However, there are some problems that the authors need to address before it is accepted for publication.

Main comments: 1. The authors did not construct a transfer function between MBT'5ME and temperature, as they claim their proxy likely reflect September temperature. I suggest the authors try to construct a transfer function to show the temperature variation quantitatively. 2. The authors compared their temperature reconstruction with pollen, hydrogen isotope and other records. The authors better explain difference between September T and pollen-inferred T. If they represent T variation in different seasons„ why do they show similar variation? 3. The authors attributed different trends in reconstructed T and measured T at the Basin Pond to Rotenone treatment. It seems that the Rotenone affected the algal community. What would the changes in algal community affect the bacteria? If bacterial community changed, why the proxy did not reflect temperature? In this case, why the proxy MBT'5ME would reflect over the past 900 years. The examples that the author listed in Section 5.1 were all from surface sediment. Were they affected by anthropogenic activities? Overall, it seems to me that the interpretation is not convincing.

I wish the authors address the concerns in revision.

---

## Author Comment (AC1) · 30 Apr 2018

We would like to thank Dr. Juzhi Hou for the insightful comments on our manuscript. We feel these comments will aid in creating a more well-rounded, thorough publication, and will be addressing these comments in an updated manuscript. One important aspect we would like to note is that comment #1 is already addressed in the supplementary information provided with the manuscript. Although a temperature to MBT transfer function would be ideal, creating a temperature calibration based off of 4 data points (see supplement) would be dubious. However, this point is valid, and we will add

a statement to the manuscript that addresses the promise our data hold for creating such a calibration if more measurements are obtained in the future. We will more explicitly address comment #2 in the revised manuscript, and will clarify the differences and similarities between seasonal temperature reconstructions. Finally, comment #3 regarding the impacts of anthropogenic disturbance on brGDGT-producing communities highlights a question that we did not thoroughly address in the manuscript. In the revision, we will add in a section directly discussing this.

---

## Referee Comment (RC2) · Anonymous Referee #2 · 27 May 2018

The authors of this paper examined the sources of brGDGTs in a small lake in Maine USA, by comparing the distribution of these compounds in (i) soils around the lake, (ii) suspended particulate matter collected at different seasons and water depths and (iii) surficial sediments. BrGDGTs were shown to be mainly produced in situ, preferentially in fall. Then, brGDGTs were analysed in a sediment core covering the last 900 yrs. Temperature estimates inferred from brGDGTs were interpreted using available paleoclimate data in the region.

This is an interesting and comprehensive study based on organic compounds increas-

ingly used as temperature proxies in continental settings. Nevertheless, some improvements are required before publication:

- The interpretation of the brGDGT data along the sedimentary core is only based on the MBT5Me index. Absolute temperatures based on the calibration developed by Sun et al. (2011) or Russell et al. (2018) are comparable (difference of ca. 1 °C, within the range of the uncertainty associated with brGDGT calibration) and should be provided in the main text, all the more as it does not change the interpretation of the data.

- I would be very cautious about the preliminary calibration between MBT derived from SPM samples and temperature, as it is based on only 4 samples.

- A local calibration between seasonal temperature (fall temperature, when brGDGTs are preferentially produced in this New England lake) and brGDGT distribution should be developed and used for temperature reconstruction along the sedimentary core.

- The discussion section, especially the comparison of the present record with other regional ones, is sometimes difficult to follow, as some explanations are missing and all the data necessary for the understanding of the reasoning are not presented. I recommend a more careful and more detailed interpretation of the data.

- Several lipid biomarkers were analyzed in the lacustrine sediments and revealed some variations in the lake productivity over time, related to anthropogenic influence. Nevertheless, the authors should better discuss the potential human impact on the brGDGT signal. Disentangling natural and anthropogenic signals in lacustrine records is a key question which should be addressed.

Other detailed comments are given below.

Line 18. Late fall rather than early spring.

Line 20. De Jonge et al., 2014 instead of 2013.

[Figure]

Lines 25-30. Please also add some recent papers examining the distribution of 5- and 6-methyl brGDGTs in lakes : Russell et al., 2018 and Dang et al., 2018 both in Organic Geochemistry.

Line 21. How long were the cores kept refrigerated? What about the evolution of organic matter (and especially brGDGTs) during storage?

Line 29. All samplings were performed after a period of 28 to 40 days, except the last point. What is the reason for such a long accumulation time (264 days)?

Lines 5-6. These 5 dates should be specified once again in the present manuscript to make easier the reading of the manuscript.

Line 9. According to Fig. 1, 20 soil samples were collected around the lake, which is not consistent with the number given in the text. This should be corrected.

Lines 14-17. Why were the TLE from SPM/soils on the one hand and lake sediments on the other hand separated differently?

Line 19. Please specify here the different algal biomarkers which will be analysed.

Lines 21-26. Please specify if the GC injections were made in split or splitless mode.

Lines 1-10. A short introduction of 5- and 6-methyl brGDGTs and the related indices should be added. Were some samples injected in duplicate/ triplicate? What is the analytical uncertainty on the MBT, CBT and IR indices?

Lines 20-25. The relative abundances of brGDGTs in all the samples (soils, SPM, lake sediments) should be given in a supplementary table. Acyclic and cyclic brGDGTs (Ia, Ib and Ic; IIa, IIb and IIc; IIIa, IIIb and IIIc) cannot be distinguished in Fig. 2. The

figure should be modified to take this comment into account. The different types of green are difficult to distinguish in Fig. 2. Please choose more contrasting colours. The brGDGT distributions are not consistent through the four collection periods: the relative abundance of GDGT I decreases from June 2014 to January 2015, in contrast with brGDGT III.

Lines 4-5. These sentences are redundant with those in lines 1-3.

Lines 24. These concentrations are not present either in a Fig. or a Table. Please provide the bulk data as Supplementary material.

Lines 27-29. These multidecadal events are difficult to distinguish.

Line 1. Please add also the recent paper by Russell et al. (2018) to the list of brGDGT lacustrine calibration.

Lines 9-13. The different arguments provided here are not convincing. The recent lacustrine calibration by Russell et al. (2018) could be applied to the NE US lakes, with all the caution needed (not the same region, difference in terms of stratification/mixing etc.). This is indeed the only calibration based on the recent analytical method proposed by Hopmans et al. (2016) allowing the separation of 5- and 6-methyl brGDGTs. Furthermore, as shown in Supp. information, the temperature variations inferred from the Russell et al. calibration are similar to those derived from the calibration by Sun et al. (2011). Therefore, the different calibrations provide different absolute temperatures (still very close, ca. 1 °C difference) but similar trends.

Line 18. Only group I and III brGDGTs can be distinguished, not the individual compounds. CBT'5Me and MBT'5Me were inverted in Fig. 6.

Line 22. Lower instead of higher.

[Figure]

Lines 1-6. This is redundant with the sentences above.

Lines 6-10. In addition to the brGDGT distribution, the brGDGT concentrations in soils, lake and SPM should also be compared.

Lines 11-12. I would be very cautious about the MBT'5Me/temperature local calibration, as it is based on only 4 points.

Lines 13-14. Where are the water temperature data?

Lines 25-26. Such a calibration should have been developed in the present study and then applied to the downcore brGDGT reconstruction.

Line 31. These concentrations should be provided as Supp. Material.

Line 33. Please remove the last sentence which is not useful.

Line 9. Please be more explicit about similarities and differences between pollen and MBT'5Me reconstructions.

Line 16. Please provide some references here. Are hydrogen isotopes from leaf waxes not mainly used are hydrological proxies?

Lines 16-21. This paragraph should be more developed. How are temperatures reconstructed from delta D of leaf waxes?

Line 28. Where are this information derived from? Fig. 7d?

Lines 29-32. The conclusion about the predominant human impact on fires remains speculative and is difficult to apprehend.

Lines 13-15. The similarities between Basin Pond reconstruction and other northern

hemisphere reconstructions are difficult to visualize.

Line 2. What do the authors mean by "not a strong cross correlation? Please provide r and p values. Lines 13-14. This trend is difficult to visualize.

Lines 17-18. What is the interest of presenting local data since the authors consider them as inaccurate? I would only present statewide trends.

Lines 24-27. This is a little too short. What are exactly the mechanisms which could explain the lower MBT'5Me values in surficial sediments? Some of these "mechanisms" may be lake-/region-dependent.

Lines 4-16 / Fig. 9a. The different algal biomarkers are difficult to distinguish in Fig. 9a. Please use contrasting colours for each biomarker.

Supplementary information

Line 17. Where is Table S1?

Line 27. Where are the fall measurements?

A MBT'5Me-temperature calibration should have been developed in the present study. The correlation presented in Fig. S1, based on only 4 scattered points, is not reliable.

---

## Author Comment (AC2) · 25 Jun 2018

We would like to thank Dr. J. Hou for the insightful comments on our manuscript. We feel these comments will aid in creating a more well-rounded publication, and we will address these comments in an revised manuscript.

Miller et al. presented a well-designed experiment to investigate the seasonality of brGDGT proxies in this paper, which is helpful to understand the mechanism of the potential temperature proxy. The authors reconstructed temperature variation in the past

900 years and suggested they could differentiate anthropogenic and natural changes. I think it is a good try to understand the seasonality of brGDGT proxies, which is worthy to be published. However, there are some problems that the authors need to address before it is accepted for publication. Main comments:

1. The authors did not construct a transfer function between MBT'5ME and temperature, as they claim their proxy likely reflect September temperature. I suggest the authors try to construct a transfer function to show the temperature variation quantitatively.

This is addressed in the supplementary information provided with the manuscript. Although a temperature to MBT transfer function would be ideal, creating a temperature calibration based on 4 data points (see supplement) would be dubious. Upon further reflection and taking into consideration the comments of referee #2, we decided it would not be appropriate to discuss this in the manuscript text. This point is valid, but it is beyond the scope of this manuscript as it cannot be adequately addressed with the currently available data.

2. The authors compared their temperature reconstruction with pollen, hydrogen isotope and other records. The authors better explain difference between September T and pollen-inferred T. If they represent T variation in different seasons, why do they show similar variation?

In response to this point as well as to a comment by Referee #2, we will clarify the differences and similarities between temperature reconstructions based on pollen and brGDGTs (Page 9 Lines 4-6). We agree that the discussion of different proxy seasonality in this section was confusing, so we will clarify this discussion. In general, summer and fall temperature fluctuations share some variation, and the differences in the proxy reconstructions presented in Figure 7 are unlikely to be solely due to differences in the exact timing of proxy production. Rather, the differences are likely attributable to discrepancies between the age models and resolution of the published records. We

will update the text to reflect this.

3. The authors attributed different trends in reconstructed T and measured T at the Basin Pond to Rotenone treatment. It seems that the Rotenone affected the algal community. What would the changes in algal community affect the bacteria?

We show reconstructions both for algal biomarkers and for higher plant biomarkers (ß-sitosterol/stanol). Our intent was to show a widespread response of the biota in Basin Pond to the 1955 Rotenone treatment. We will add text to speculate on the response of brGDGT producers in Basin Pond to the 1955 treatment. However, it is unclear to us how this treatment may have affected bacteria, and to the best of our knowledge, little work has been done to investigate this. Because other aspects of lake productivity changed, we speculate that this may have affected the microbial community of Basin Pond, including brGDGT producers. Further work to quantify the impacts of anthropogenic changes on brGDGT producers, as well as more work to identify brGDGT producers (which remain unknown), is needed to better understand these relationships, which is beyond the scope of this manuscript.

If bacterial community changed, why the proxy did not reflect temperature? In this case, why the proxy MBT'5ME would reflect over the past 900 years.

This is a good point, but we feel it is a bit of an oversimplification of our argument. The veracity of our 900-year brGDGT-based temperature reconstruction is supported by its comparison to other paleorecords, and the clear relationship between MBT'5Me and local temperature measured at Basin Pond. However, if we focus on simply the last 100 years, we find a surprising cooling since 1970, in contrast to instrumental records of temperature change for the state of Maine (Figure 9). We will add text detailing a few possible mechanisms for this effect (Page 11, Lines 20-22). We note that all of these mechanisms may be affected by changes to the Basin Pond ecosystem initiated by the addition of Rotenone in 1955. We note that any post-depositional changes occur in the upper ∼3.5 cms of the sediment column, and that the 900-year brGDGT-based

temperature reconstruction would not have been altered by anthropogenic impacts. This is supported by our comparison of the brGDGT-based record to other regional records, as well as by other published data (i.e. the age model for the Basin Pond cores BP2014-5D, BP2014-3D, which show no 14C age reversals) (Miller et al. 2017).

The examples that the author listed in Section 5.1 were all from surface sediment. Were they affected by anthropogenic activities?

The answer to this question is location-specific and varies regionally in terms of the type and extent of anthropogenic influence for each of the records/studies mentioned in this section. It is beyond the scope of this study to address this question and revisit all of the existing calibrations for signs of possible anthropogenic influence. Nevertheless, this is an interesting question and merits study in the future.

Overall, it seems to me that the interpretation is not convincing.

We hope that our responses to Referee #1's comments have clarified and supported our interpretation. We will modify Section 5.3 so that it explicitly and clearly lays out our interpretation of the 900-year brGDGT-based record. We also note that while uncertainties to some aspects of the interpretation exist and may not be conclusively resolved at this time, our brGDGT and algal biomarker records from Basin Pond add a new and high-resolution record to the available paleoclimate records. In particular, there are relatively few land temperature reconstructions from Maine and our study helps to fill this gap. It is likely that as the brGDGT temperature proxy becomes better understood, that interpretation of our Basin Pond record, as well as other previously published brGDGT records, will evolve accordingly.

I wish the authors address the concerns in revision.

Referee #1's concerns will be addressed in the revision.

---

## Author Comment (AC3) · 25 Jun 2018

We would like to thank Referee #2 for their suggestions and detailed review of our manuscript. Referee #2 requested several revisions, primarily within the discussion section. One of the main issues that referee #2 raised was the lack of applying a current published temperature calibration to the MBT'5me record. After consideration, we have agreed to provide an additional plot of our Basin Pond MBT'5me record with two different temperature calibrations (Dang et al. 2018, and Russell et al. 2018) that were developed using analytical methods directly comparable to our own, that allow for

the separation of 5me and 6me brGDGTs during HPLC analysis. Referee #2 stated that we should develop a calibration of our brGDGT observations to local temperatures, and also noted that we do not have enough data for such a calibration to be robust. We would like to emphasize that the goal of this study was not to provide a brGDGT to temperature calibration for Basin Pond. Rather, the goal of this study was to understand the timing and depth profile of brGDGT production in Basin Pond in order to address potential seasonal bias in the 900-year climate reconstruction. We agree with referees #1 and #2 in their assertions that a focused effort to develop a local calibration would be a useful contribution to our understanding of brGDGTs. Although this is beyond the scope of the main manuscript, our data have been plotted against local temperature in the supplementary materials, as we believe future efforts to develop a local calibration may benefit from these data. The main purpose of including this in the supplement was to highlight the potential Basin Pond holds in producing a temperature calibration in the future.

All other comments have been addressed in line below.

- The interpretation of the brGDGT data along the sedimentary core is only based on the MBT5Me index. Absolute temperatures based on the calibration developed by Sun et al. (2011) or Russell et al. (2018) are comparable (difference of ca. 1 ǓęC, within the range of the uncertainty associated with brGDGT calibration) and should be provided in the main text, all the more as it does not change the interpretation of the data.

We present our data as MBT'5Me values rather than applying a temperature calibration because the two existing calibrations based on methods that separate 5me and 6me brGDGTs were developed from lakes located in very different regions (China and Africa) (Dang et al. 2018, Russell et al. 2018) than Basin Pond (NE USA). Furthermore, brGDGT distributions in Basin Pond differ significantly from those of the Chinese and African Lakes (we will add a new ternary diagram plotting the hexa-, penta-, and tetra-methylated brGDGTs to clearly illustrate this- see attached). For this reason, these

calibrations should be treated with caution when applied to Basin Pond sediments to reconstruct temperature. However, based on the request of the referee, we will plot our data on these 2 different temperature calibrations (Dang et al. 2018, Russell et al. 2018) in an additional figure that will be provided in the main text of the manuscript, while also continuing to emphasize our reasons for exercising caution when using these calibrations at Basin Pond. To summarize, our data will be presented as 1) an additional figure with MBT'5me, growth temp (Dang et al. 2018), and mean annual air temp (Russell et al. 2018), 2) a ternary diagram (see attached figure) in the supplementary information showing that the Basin Pond samples and the Chinese and African Lakes samples are characterized by different brGDGT distributions,, and 3) as MBT'5me and deviation from mean, on figures 7 and 8. Other temperature calibrations mentioned by the referee (e.g., Sun et al. 2011) are based on different analytical methods that did not fully separate 6Me isomers, and are therefore not suitable calibrations to apply to our MBT'5me record.

- I would be very cautious about the preliminary calibration between MBT derived from SPM samples and temperature, as it is based on only 4 samples.

We agree - because we are also very cautious about this preliminary calibration based on very few data points, we placed it in the supplement rather than the main text. We will revise the main text to better emphasize this important point.

- A local calibration between seasonal temperature (fall temperature, when brGDGTs are preferentially produced in this New England lake) and brGDGT distribution should be developed and used for temperature reconstruction along the sedimentary core.

We agree and this study in an area of our ongoing research. However, this is beyond the scope of the presented work. Here, the main focus in on presenting our new 900-year reconstruction.

- The discussion section, especially the comparison of the present record with other regional ones, is sometimes difficult to follow, as some explanations are missing and

all the data necessary for the understanding of the reasoning are not presented. I recommend a more careful and more detailed interpretation of the data.

We agree and thank the referee for this suggestion, and will revise the discussion text to make it easier to follow, more thorough, and more detailed. Changes to the text to address this point include: revision of Section 5.3 through 5.8.

- Several lipid biomarkers were analyzed in the lacustrine sediments and revealed some variations in the lake productivity over time, related to anthropogenic influence. Nevertheless, the authors should better discuss the potential human impact on the brGDGT signal.

We agree, and the potential human impact on the brGDGT signal will be discussed in more detail on Page 12, Lines 10–12. We have also addressed this in our response to Referee #1. However, we note that there is a general lack of knowledge on brGDGT producers, we therefore do not wish to speculate too much on this subject.

Disentangling natural and anthropogenic signals in lacustrine records is a key question which should be addressed.

We note the potential anthropogenic influence on the upper 3.5 cm of our reconstruction following the addition of Rotenone to the lake in 1955. However, the majority of our 900-year record is not subject to this kind of anthropogenic disruption. We will add text detailing anthropogenic influence in Section 2.1. We note that the precise impact of any kind of anthropogenic disturbance on brGDGT reconstructions is hindered by an incomplete understanding of the organisms responsible for brGDGT production in general. We leave this to future work in the brGDGT scientific community.

Other detailed comments are given below.

Line 18. Late fall rather than early spring.

fixed.

Line 20. De Jonge et al., 2014 instead of 2013.

fixed.

Lines 25-30. Please also add some recent papers examining the distribution of 5- and 6-methyl brGDGTs in lakes: Russell et al., 2018 and Dang et al., 2018 both in Organic Geochemistry.

These references, which were published since the first draft of this paper was generated, will be added .

Line 21. How long were the cores kept refrigerated? What about the evolution of organic matter (and especially brGDGTs) during storage?

The cores were collected in March 2014 (Page 3, line 16) and were stored unsplit and refrigerated (4°C, in the dark) for one month prior to subsampling (this information will be added to Page 3, lines 19-20). Although growth and alteration of the brGDGT signal during storage is a potentially important form of diagenesis for s, the sample processing in question is relatively rapid compared to some other existing studies that have utilized brDGTs (i.e. these cores were not stored for years prior to subsampling). This information will be added to the manuscript. We note that while this point should be investigated by future studies, many previous brGDGT studies have been done on old sediments and this is not believed to be a significant factor influencing the brGDT records (e.g., Weijers et al. 2007).

Line 29. All samplings were performed after a period of 28 to 40 days, except the last point. What is the reason for such a long accumulation time (264 days)?

The sediment traps, which were designed and constructed at UMass, required us to enter the water to pull up the traps and therefore were only retrieved and samples

collected while the lake was ice-free. During the winter we were not able to pull out the sediment traps because entering the water was not possible; additionally, cutting numerous large holes into the ice would have presented a safety hazard for the people utilizing Basin Pond for winter recreation. .

Lines 5-6. These 5 dates should be specified once again in the present manuscript to make easier the reading of the manuscript.

We are unsure what the referee meant by this. Nevertheless we will make it clear when sampling took place and will repeat the sampling dates again further in the updated manuscript.

Line 9. According to Fig. 1, 20 soil samples were collected around the lake, which is not consistent with the number given in the text. This should be corrected.

Thank you- this is now correct, and the number of soil samples (10 samples) we analyzed matches the number of samples in Fig. 1.

Lines 14-17. Why were the TLE from SPM/soils on the one hand and lake sediments on the other hand separated differently?

For the purposes of this study, we are only interested in compounds that eluted in the apolar and polar fractions. Another study using the ketone fraction of the sediment core has already been published (Miller et al., 2017) and initially was designed to look for lacustrine alkenones (but they were not present). The SPM/soil samples were measured solely for this study, after we determined separating out the ketone fraction was unnecessary. However, we note that our separation procedure does not omit any compounds. For the SPM/soil samples, any compounds that would have eluted in the ketone fraction just end up in the polar fraction.

Line 19. Please specify here the different algal biomarkers which will be analyzed.

This will be added.

Lines 21-26. Please specify if the GC injections were made in split or splitless mode.

Injections were made in splitless mode. This will be added to the text.

Lines 1-10. A short introduction of 5- and 6-methyl brGDGTs and the related indices should be added. Were some samples injected in duplicate/ triplicate? What is the analytical uncertainty on the MBT, CBT and IR indices?

We will include an introduction to 5- and 6-methyl brGDGTs on Page 2, Lines 19-24. In addition, section 4.4 "BrGDGT isomer ratios" also deals with this topic. Yes, a subset of the samples were analyzed in duplicate. Analytical uncertainty will be added in this section as well.

Lines 20-25. The relative abundances of brGDGTs in all the samples (soils, SPM, lake sediments) should be given in a supplementary table.

We plan to make these data publically available. All Basin Pond data presented here will be archived at the NOAA Paleoclimate Data Center and we plan to submit this data as soon as the manuscript is accepted for publication. If the editor wishes, we can add data tables as supplementary materials as well.

Acyclic and cyclic brGDGTs (Ia, Ib and Ic; IIa, IIb and IIc; IIIa, IIIb and IIIc) cannot be distinguished in Fig. 2. The figure should be modified to take this comment into account.

This manuscript does not explicitly discuss the differences in abundances of the a b, and c compounds (note also that relative abundances of the b and c compounds are very small; see Page 5 Lines 22–23). Therefore, we chose not to display them in Figure 2. Interested parties can access the a, b, and c data in the included data with the NOAA Paleoclimate Data Center upon publication.

The different types of green are difficult to distinguish in Fig. 2. Please choose more contrasting colours.

We will update this figure with more contrasting colors.

The brGDGT distributions are not consistent through the four collection periods: the relative abundance of GDGT I decreases from June 2014 to January 2015, in contrast with brGDGT III.

We made this observation on Page 5, lines 23-25: However, in June and July 2014, Group I brGDGTs were the most abundant, whereas in September 2014 and January 2015, reductions in Group I brGDGTs were accompanied by increases in Group III brGDGTs. We will remove the sentence about distributions being consistent through time, and add additional sentences that accurately describe the changes.

Lines 4-5. These sentences are redundant with those in lines 1-3.

This paragraph will be rearranged for clarity and to reduce redundancy. However, we note that we separately discuss brGDGT fluxes and brGDGT distributions in this paragraph; while it may seem like there is redundancy, we find upon careful reading of these lines that all of the information presented is new and necessary.

Lines 24. These concentrations are not present either in a Fig. or a Table. Please provide the bulk data as Supplementary material.

The data will be provided upon publication with the NOAA Paleoclimate Data Center, or if the editor wishes a supplementary table.

Lines 27-29. These multidecadal events are difficult to distinguish.

We will distinguish the events by highlighting them with arrows in fig 7.

Line 1. Please add also the recent paper by Russell et al. (2018) to the list of brGDGT lacustrine calibration.

The first sentences of the paragraph are discussing older lakes calibrations. The Russell calibration for African lakes is discussed and cited later in the paragraph after discussion of the new separation method for 5- and 6-methyl isomers.

Lines 9-13. The different arguments provided here are not convincing. The recent lacustrine calibration by Russell et al. (2018) could be applied to the NE US lakes, with all the caution needed (not the same region, difference in terms of stratification/mixing etc.). This is indeed the only calibration based on the recent analytical method proposed by Hopmans et al. (2016) allowing the separation of 5- and 6-methyl brGDGTs. Furthermore, as shown in Supp. information, the temperature variations inferred from the Russell et al. calibration are similar to those derived from the calibration by Sun et al. (2011). Therefore, the different calibrations provide different absolute temperatures (still very close, ca. 1 âŮęC difference) but similar trends.

As mentioned previously, we will present our data in a new figure in the manuscript with the Russell et al. 2018 African Lakes calibration, the Dang et al. 2018 Chinese Lakes calibration, as these are the only calibrations based on the recent analytical method, by Hopmans et al. 2016, that we used in this study. However, as mentioned previously, we also will add a ternary diagram to the supplementary information, which shows that brGDGT distributions of the samples in this study fall outside those of the African and Chinese Lake samples. This means the brGDGT producers in Basin Pond, and their sensitivity to temperature, are potentially different from those in the lakes studied by Russel et al. 2018 or Dang et al. 2018. We therefore wish to emphasize that presenting the data in this way requires caution, as it invites a suite of misinterpretations for readers who are not familiar with the proxy. Because of this, we will continue to present our data as MBT'5Me values in Figures 7 and 8, and indicate with words and colors in these figures that higher MBT'5Me values indicate warmer temperatures, and vice versa.

Line 18. Only group I and III brGDGTs can be distinguished, not the individual compounds.

We will correct the text to reflect this. We note that the b and c compounds make up <2% of the relative abundance in our samples. CBT'5Me and MBT'5Me were inverted in Fig. 6. This will be fixed.

Line 22. Lower instead of higher.

This will be fixed.

Lines 1-6. This is redundant with the sentences above.

We will edit this paragraph to eliminate redundancy.

Lines 6-10. In addition to the brGDGT distribution, the brGDGT concentrations in soils, lake and SPM should also be compared.

brGDGT fluxes are explicitly discussed with reference to the SPM samples (e.g. Page 5 Lines 25-27 , Figure 3). brGDGT concentrations in soils (Page 5 Lines 20) and sediments (Page 6 Lines 24) are also noted. These quantities do not have the same units and are thus not directly comparable; furthermore, brGDGT concentrations have a wide range of values in the literature, making it difficult to interpret how the soil, sediment, and SPM concentrations compare and how they represent relevant environmental parameters.

Lines 11-12. I would be very cautious about the MBT'5Me/temperature local calibration, as it is based on only 4 points.

This reasoning is why we put it in the supplement, and noted that future work should address this promising but presently thin calibration.

Lines 13-14. Where are the water temperature data?

Water temperature data are presented in Figure 4(a), and we will add a citation (Frost, 2005) which details the yearly cycle of water temperatures as a function of depth in Basin Pond.

Lines 25-26. Such a calibration should have been developed in the present study and then applied to the downcore brGDGT reconstruction.

We will remove this sentence. Because of the limited number of SPM samples we were able to collect for this study, we recognize that it is not a calibration experiment, and should not be considered as such. Instead it is a detailed assessment of seasonal production biases coupled with the downcore application of a powerful emerging paleotemperature proxy, and the main focus of this paper is on the downcore record.

Line 31. These concentrations should be provided as Supp. Material.

These concentrations will be provided in the dataset with the NOAA Paleoclimate Data Center upon acceptance for publication.

Line 33. Please remove the last sentence which is not useful.

This sentence will be removed.

Line 9. Please be more explicit about similarities and differences between pollen and MBT'5Me reconstructions.

This will be addressed in the text in the updated manuscript (see response to referee #1 comments).

Line 16. Please provide some references here.

We will add a reference to the review by Sachse et al., 2012. Are hydrogen isotopes from leaf waxes not mainly used are hydrological proxies? Hydrogen isotopes of leaf waxes can reflectchanges in temperature (e.g. high-latitude sites) or changes in hydrological processes (e.g. tropical sites), or changes in the dominant moisture source, and interpretation is dependent on site-specific parameters. This is addressed in the text but the details of these processes are irrelevant to this study - we provide the dD record from Gao et al. 2017, which the authors interpret as a temperature record, because it is one of the few available records from New England.

Lines 16-21. This paragraph should be more developed. How are temperatures reconstructed from delta D of leaf waxes?

This is beyond the scope of this study. We provide these data simply for comparison to our own record. For more information, see the study of Gao et al., 2017. However, we will make it clear in the manuscript text that Gao et al. (2017) interpret their dD record as a temperature proxy.

Line 28. Where are this information derived from? Fig. 7d?

This information is derived from comparison between Figure 7g and Figure 7d–f. This will be updated in the text.

Lines 29-32. The conclusion about the predominant human impact on fires remains speculative and is difficult to apprehend.

We will remove the speculative sentence about the causes of fire changing through time. Our reasoning was based on the fact that the regional records indicate the climate has generally been getting cooler (Figure 7a–c) and wetter (Figure 7d–f) over the last 900 years, whereas the fire record indicates higher charcoal counts in the last 200 years compared to the previous 700. We note this is beyond the scope of the present study and as the referee noted highly speculative so we will remove it from the text.

Lines 13-15. The similarities between Basin Pond reconstruction and other northern hemisphere reconstructions are difficult to visualize.

We will make the text more explicit about the similarities and differences between the Basin Pond reconstruction and northern hemisphere reconstructions. Additionally we will revise Figure 8 to better facilitate these comparisons.

Line 2. What do the authors mean by "not a strong cross correlation? Please provide r and p values.

Fixed, r and p values will be added to text.

Lines 13-14. This trend is difficult to visualize.

We are unsure exactly which trend the referee is talking about, but all the trends discussed are shown in Figure 9. This paragraph will be edited to focus on the trends that we are showing, and to remove superfluous information about other trends that are not relevant for the present discussion.

Lines 17-18. What is the interest of presenting local data since the authors consider them as inaccurate? I would only present statewide trends.

We agree. We will adjust the text and Figure 9 accordingly.

Lines 24-27. This is a little too short. What are exactly the mechanisms which could explain the lower MBT'5Me values in surficial sediments?

We will add a sentence which lists possible mechanisms to explain this trend without speculating too much about our own data. Furthermore, we will cite other studies showing a similar trend in the uppermost sediments, and that the mechanism causing this is not currently well understood.

Some of these "mechanisms" may be lake-/region-dependent.

We agree.

Lines 4-16 / Fig. 9a. The different algal biomarkers are difficult to distinguish in Fig. 9a. Please use contrasting colours for each biomarker.

In order to present data in a color-blind friendly manner, we decided to stick with shades of gray. However, we agree with the referee that the way it was arranged made it difficult to distinguish the colors and link them to the respective biomarkers. We will change this figure to make it more reader-friendly in response to the referee's request.

Supplementary information

Line 17. Where is Table S1?

Table S1 is the first table in the supplement, but was mislabeled as S2. This will be fixed.

Line 27. Where are the fall measurements?

The average measured temperatures for each of the sediment trap collection periods are listed in Table S2.

A MBT'5Me-temperature calibration should have been developed in the present study.

While this is a fantastic goal, we found that a correlation based on only four scattered points would be spurious (see following comment). However, we note that the main goal of this study was to reconstruct temperature trends over the past 900 years.

The correlation presented in Fig. S1, based on only 4 scattered points, is not reliable.

This was our reasoning as to why we chose to include it in the supplement and not actually apply it to the data in the main text.

[Figure]

[Figure]

**Fig. 1.**

---

## Author Response (AR1)

September 12, 2018

Dear Editor Rousseau,

We are re-submitting the revised version of the manuscript "*A 900-year New England temperature reconstruction from* in situ*, seasonally produced branched glycerol dialkyl glycerol tetraethers (brGDGTs)*" for publication in *Climate of the Past*. Two referees, J. Hou and Anonymous, provided helpful feedback and comments that we utilized to improve the manuscript. Per their requests, major adjustments were made to the manuscript, particularly in the discussion section, to increase clarification and facilitate easier interpretation of our study. The major changes include: 1) the addition of two figures (Figures 7 & 8) to better clarify differences in brGDGT distributions between Basin Pond and other studies, 2) an expanded and re-organized Discussion section, addressing the utility of two published brGDGT calibrations using the same UHPLC method we used here (Section 5.2), and 3) the potential for human impacts on the brGDGT signal since the 1950s (Section 5.7). In addition, numerous minor revisions helped improve the readability of the manuscript and the figures. Overall, we feel that these revisions have resulted in a substantially improved manuscript.

The major critique both reviewers had of our work was the presentation of our data in terms of the MBT'$_{5ME}$ index, rather than as an inferred temperature based on applying a published calibration. The two new figures (Figures 7 and 8) and a section in the Discussion (Section 5.2) have been added to address this. We note that while the two published brGDGT temperature calibrations of Russell et al. (2018) and Dang et al. (2018) show similar trends over the last 900 years, they differ in the expression of decadal- and centennial-scale variability. Furthermore, we now show that Basin Pond brGDGTs have a different distribution than those measured at other locations around the globe, underlining the fact that a local calibration is likely needed to provide the accurate temperature reconstructions. Although the current study has laid the groundwork for this (i.e. Supp. Fig. S1) the development of a proper local calibration is beyond the scope of this study, which is mainly focused on reconstructing climate over the past 900 years.

Overall, this work is important because it contributes to our knowledge of Northeast US Late Holocene climate using a brGDGT-based temperature proxy. Our interpretations are strengthened by detailed, *in situ* assessment of the production, transport, and deposition of brGDGTs in Basin Pond. For this reason, we feel our study is of interest to both the paleoclimate and proxy-development communities and is thus especially suited for *Climate of the Past*. It is clear from both reviewer comments that they fundamentally agree about the importance and potential impact of this work, and we believe that the changes we have made following their suggestions have made our study even stronger. We wish to sincerely thank them for their thoughtful reviews!

A document detailing the specific changes made in response to the comments of both reviewers has been provided along with the revised manuscript, and a manuscript with track changes. We look forward to your decision regarding our manuscript!

Sincerely,

Daniel R. Miller (on behalf of all co-authors)

**Referee #1 J. Hou**

**We would like to thank Dr. J. Hou for the insightful comments on our manuscript. We feel these comments have aided in creating a more well-rounded publication, and we have addressed these comments in the revised manuscript.**

Miller et al. presented a well-designed experiment to investigate the seasonality of brGDGT proxies in this paper, which is helpful to understand the mechanism of the potential temperature proxy. The authors reconstructed temperature variation in the past 900 years and suggested they could differentiate anthropogenic and natural changes. I think it is a good try to understand the seasonality of brGDGT proxies, which is worthy to be published. However, there are some problems that the authors need to address before it is accepted for publication. Main comments:

1. The authors did not construct a transfer function between MBT'5ME and temperature, as they claim their proxy likely reflect September temperature. I suggest the authors try to construct a transfer function to show the temperature variation quantitatively.
**This is addressed in the supplementary information provided with the manuscript. Although a temperature to MBT'5Me transfer function would be ideal, creating a temperature calibration based on four data points (see supplement), from our four SPM sampling dates, would be dubious. To help address this point, we have added a new section to the discussion (section 5.2) along with two new figures (7 and 8) comparing our MBT'5Me record to published temperature calibrations. However, constructing a regional temperature calibration is beyond the scope of this manuscript.**

2. The authors compared their temperature reconstruction with pollen, hydrogen isotope and other records. The authors better explain difference between September T and pollen-inferred T. If they represent T variation in different seasons, why do they show similar variation?
**In response to a comment by Referee #2, we have clarified the differences and similarities between temperature reconstructions based on pollen and brGDGTS and possible mechanisms for the differences in proxies (Page 13 Lines 19 – 24). In general, summer and fall temperature fluctuations share some variation, and the differences in the proxy reconstructions presented in Figure 9 are unlikely to be solely due to differences in the exact timing of proxy production. Rather, some of these differences are likely attributable to discrepancies between the age models and the sampling resolution of the published records. We have updated the text to reflect this (Page 13 Lines 19 – 24).**

3. The authors attributed different trends in reconstructed T and measured T at the Basin Pond to Rotenone treatment. It seems that the Rotenone affected the algal community. What would the changes in algal community affect the bacteria? **Figure 11 shows relative abundances of both algal biomarkers and higher plant biomarkers (ß-sitosterol/stanol) to illustrate a widespread response of the biota in Basin Pond to the 1955 Rotenone treatment. We have added text (section 4.6; page 17 line 17 – page 18 line 2) to speculate on the response of brGDGT producers in Basin Pond to this treatment. However, it is unclear howrotenone treatment may have affected brGDGT-producing bacteria, which do not exhibit any clear changes in concentration before/after treatment. Because other aspects of lake productivity changed,**

we speculate that this may have affected the Basin Pond microbial community , including brGDGT producers. We have added a few sentences at Page 17, Line 29 – Page 18, line 2 to better clarify this point. Further work to quantify the impacts of anthropogenic changes on brGDGT producers, as well as to identify brGDGT producers (which remain unknown), is needed to better understand these relationships.

If bacterial community changed, why the proxy did not reflect temperature? In this case, why the proxy MBT'5ME would reflect over the past 900 years. **This is a good point, and its one that has generated much discussion. The veracity of our 900-year brGDGT-based temperature reconstruction is supported by its comparison to other paleorecords, and the clear relationship between MBT'$_{5ME}$ and local temperature measured at Basin Pond. However, if we focus on simply the last 100 years, we observe cooling since 1970, in contrast to instrumental records of temperature change for the state of Maine (Figure 11). We have added much text in Section 5.6 detailing a few possible mechanisms for this effect (Page 16, Lines 19 – Page 17, Line 16). We note that all of these mechanisms may be affected by changes to the Basin Pond ecosystem initiated by the addition of Rotenone in 1955, which could have affected only the upper ~3.5 cm of the sediment column. Thus, the great majority of the 900-year brGDGT-based temperature reconstruction likely would not have been altered by anthropogenic impacts. This is supported by our comparison of the brGDGT-based record to other regional records, as well as by other published data (i.e. the age model for the Basin Pond cores BP2014-5D and BP2014-3D, which show no [14]C age reversals) (Miller et al. 2017).**

The examples that the author listed in Section 5.1 were all from surface sediment. Were they affected by anthropogenic activities? **We have made substantial changes to the manuscript – section 5.1 now discusses Sources and seasonal production of brGDGTs, while section 5.2 addresses the downcore calibration to temperature. The answer to this question posed by the referee deals with a number of previously published studies that are location-specific and vary regionally in terms of the type and extent of anthropogenic influence. It is beyond the scope of this study to revisit all of the existing studies for signs of anthropogenic influence. In addition, we do not think that the signal in the top ~3.5 cm compromises the veracity of our 900-year reconstruction, nor do we think that potential anthropogenically-influenced overprinting of brGDGT distributions in surface sediments compromises the robust correlation between brGDGT distributions and temperature that has been noted by an array of published studies (see Sections 5.3 and 5.4). Furthermore, ongoing research to calibrate the Basin Pond brGDGTs to temperature is being performed (see Supplementary materials). Nevertheless, this is an interesting question and merits future study.**

Overall, it seems to me that *the interpretation is not convincing.* **We hope that our responses to Referee #1's comments have clarified and supported our interpretation. We have made major changes to all discussion sections (5.1, 5.2, 5.3, 5.5, and 5.6) so that it clearly lays out our interpretation of the 900-year brGDGT-based record. We also note that while uncertainties to some aspects of the interpretation exist and may not be conclusively resolved at this time, our brGDGT and algal biomarker records from Basin Pond add a new and high-resolution record to available paleoclimate records for the NE US. In particular, there are relatively few terrestrial temperature reconstructions from Maine and our study helps to fill this gap. It is likely that as the brGDGT temperature proxy becomes better understood, and**

**as new temperature calibrations are developed, the interpretation of our Basin Pond record will evolve accordingly. By presenting our data as MBT'$_{5ME}$ index values, we have made this easier for future studies ,which may revisit these data.**

I wish the authors address the concerns in revision. We have tried to address all of **Referee #1's concerns in the revision.**

**Referee #2 Anonymous**

Referee #2 requested several revisions, primarily within the discussion section. One of the main issues that referee #2 addressed was the lack of applying a current published temperature calibration to the MBT'5ME record. In response, we have provided our Basin Pond MBT'5ME record plotted with 2 different temperature calibrations (Dang et al. 2018 and Russell et al. 2018; Figure 7) that were developed using analytical methods directly comparable to our own, i.e. separating 5-methyl and 6-methyl brGDGTs during HPLC analysis (Figure 7, Discussion section 5.2). Referee #2 stated that we should develop a calibration of our brGDGT observations to local temperatures, while also noting that we do not have enough data for such a calibration to be robust. We agree that we do not have enough data for such a calibration, and we would like to emphasize that the primary goal of this study was not to provide a brGDGT to temperature calibration for Basin Pond. Rather, our goal was to (1) present a new 900 year record and (2) understand the timing and depth profile of brGDGT production in Basin Pond in order to address potential seasonal bias in the 900-year brGDGT reconstruction. We agree with the referees in their assertions that a focused effort to develop a local calibration would be a useful contribution to the paleoclimate community's understanding of brGDGTs. Although this is beyond the scope of this manuscript, our preliminary data have been plotted against local temperature in the supplementary materials (Figure S1), as future efforts to develop a local calibration should incorporate these data. The main purpose of including this in the supplement was to highlight the potential of Basin Pond for future paleoclimate reconstructions.

- The interpretation of the brGDGT data along the sedimentary core is only based on the MBT5Me index. Absolute temperatures based on the calibration developed by Sun et al. (2011) or Russell et al. (2018) are comparable (difference of ca. 1 ˚C, within the range of the uncertainty associated with brGDGT calibration) and should be provided in the main text, all the more as it does not change the interpretation of the data.

In the original manuscript, we presented our data as MBT'5ME values rather than applying a temperature calibration because the two existing calibrations based on methods that separate 5-methyl and 6-methyl brGDGTs were developed from lakes located in different regions (China and Africa) (Dang et al. 2018, Russell et al. 2018) than mid-latitude Basin Pond. We feel that these calibrations should be treated with caution when attempting to interpret absolute temperatures at Basin Pond. However, based on the request of the reviewer, we have plotted our data on these 2 different temperature calibrations (Dang et al. 2018, Russell et al. 2018) in a new figure (Figure 7), along with additional discussion (Section 5.2), in the main text of the manuscript. To summarize, this is presented in the revised manuscript as 1) an additional figure with MBT'5ME, growth temp (Dang et al. 2018), and mean annual air temp (Russell et al. 2018), and 2) a ternary plot showing that brGDGT distribution in the Basin Pond samples and African Lakes samples are distinct (Figure 8). These figures highlight that applying either previously published brGDGT temperature calibration to Basin Pond samples may be misleading, and therefore we present our data as the MBT'5ME index values in Figures 9 and 10. Other temperature calibrations mentioned by the referee (e.g., Sun et al. 2011) are based on different analytical methods that did not fully separate 6-methyl isomers (MBT/CBT proxy; Weijers et al., 2007), and are therefore not suitable calibrations to apply to our MBT'5ME record.

- I would be very cautious about the preliminary calibration between MBT derived from SPM samples and temperature, as it is based on only 4 samples.
**We agree and because we are also very cautious about this preliminary calibration based on few data points, we placed it in the supplement rather than the main text. We have re-emphasized this in the main text to better state this important point (Page 12, Line 27 – Page 13, Line 2).**

- A local calibration between seasonal temperature (fall temperature, when brGDGTs are preferentially produced in this New England lake) and brGDGT distribution should be developed and used for temperature reconstruction along the sedimentary core.
**We agree, and this is an area of our ongoing research. However, this is beyond the scope of the present work. Please see specific comments below for more details.**

- The discussion section, especially the *comparison of the present record with other regional ones*, is sometimes difficult to follow, as some explanations are missing and all the data necessary for the understanding of the reasoning are not presented. I recommend a *more careful and more detailed interpretation* of the data.
**We agree and thank the referee for this suggestion, and have revised the discussion text to make it easier to follow and more detailed. Changes to the text to address this point include several substantial changes, as well as numerous minor revisions, of the discussion section 5.3 (Page 13, Line 15 – Page 14, Line 17). These are outlined in detail below.**

- Several lipid biomarkers were analyzed in the lacustrine sediments and revealed some variations in the lake productivity over time, related to anthropogenic influence. Nevertheless, the authors should better discuss the *potential human impact on the brGDGT signal*.
**We agree, and text discussing the potential human impact on the brGDGT signal, as well as clarification of our interpretation involving anthropogenic impacts, has been added in Section 5.6.  Furthermore, we have added an additional section discussing human impact in the region (section 2.1, Page 4 Lines 19-29). We have also addressed this in our response to Referee #1. However, we note that there is a general lack of knowledge on brGDGT producers, we therefore do not wish to speculate too much on this subject. We hope that in the revised manuscript we achieved a good balance between complete discussion of the potential influences without being overly speculative.**

- Disentangling natural and anthropogenic signals in lacustrine records is a key question which should be addressed.
**We agree that this point needed better in-depth discussion. We have noted the potential anthropogenic influence on the upper 3.5 cm of our reconstruction following the addition of Rotenone to the lake in 1955 (Section 5.6, Page 17, Line 17 – Page 18, Line 2). Furthermore, we have added text detailing anthropogenic influence at Basin Pond in Section 2.1 (Page 4 Lines 19-29), and have provided evidence that the majority of our 900 year record is not subject to this kind of anthropogenic disruption seen in the 20th century (sections 5.3-5.4). Unfortunately, the precise impact of any kind of anthropogenic disturbance on brGDGT reconstructions is hindered by an incomplete understanding of the organisms responsible for**

**brGDGT production in general. We leave this to future work in the brGDGT scientific community.**

Other detailed comments are given below.

Line 18. Late fall rather than early spring.
**fixed.**

Line 20. De Jonge et al., 2014 instead of 2013.
**fixed.**

Lines 25-30. Please also add some recent papers examining the distribution of 5- and 6-methyl brGDGTs in lakes: Russell et al., 2018 and Dang et al., 2018 both in Organic Geochemistry. **These references, which have been published since the first draft of this paper was composed, have been added and discussed in detail throughout the paper.**

Line 21. How long were the cores kept refrigerated? What about the evolution of organic matter (and especially brGDGTs) during storage?
**The cores were collected in March 2014 and were stored unsplit and refrigerated in conditions similar to those at the lake floor (4ºC, in the dark) for one month prior to subsampling (this information has been added to Page 5, lines 8-9). Although growth and alteration of the brGDGT signal during storage is a potentially important form of diagenesis, the sample processing in question is relatively rapid compared to some other existing studies that have utilized brDGTs (i.e. these cores were not stored for years prior to subsampling). This core storage information has been added to the manuscript. We note that while this point should be investigated by future studies, many previous brGDGT studies have been done on old sediments and this is not believed to be a significant factor influencing the brGDT records (e.g., Weijers et al. 2007 Science).**

Line 29. All samplings were performed after a period of 28 to 40 days, except the last point. What is the reason for such a long accumulation time (264 days)?
**The sediment traps, which were designed and constructed at UMass, required us to enter the water to pull up the traps and therefore were only retrieved while the lake was ice-free. During the winter we were not able to pull out the sediment traps because entering the water was not possible. Additionally, cutting numerous large holes into the ice would have presented a safety hazard for the people utilizing Basin Pond for winter recreation and we were advised not to do this. We have added text discussing this in Section 2.3 (Page 5, Lines 18-23).**

Lines 5-6. These 5 dates should be specified once again in the present manuscript to make easier the reading of the manuscript. **We agree – we have restated these 5 dates with much more clarity in the methods (Section 2.3, Page 5, Lines 18-23) and in the discussion, and have endeavored to make it clear when sampling took place throughout the manuscript. To facilitate easiest reading of the manuscript, we refer to each sampling period as the month at the midpoint of the period occurs.**

Line 9. According to Fig. 1, 20 soil samples were collected around the lake, which is not consistent with the number given in the text. This should be corrected. **Thank you- this is now correct, and the number of soil samples (10 samples) we analyzed matches the number of samples in Fig. 1.**

Lines 14-17. Why were the TLE from SPM/soils on the one hand and lake sediments on the other hand separated differently? **For the purposes of this study, we are only interested in compounds that eluted in the apolar and polar fractions. Another study using the ketone fraction of the sediment core has already been published (Miller et al., 2017) and initially was designed to look for lacustrine alkenones (they were not present). The SPM/soil samples were measured solely for this study, after we determined separating out the ketone fraction was unnecessary. However, we note that our separation procedure does not omit any compounds. For the SPM/soil samples, any compounds that would have eluted in the ketone fraction end up in the polar fraction.**

Line 19. Please specify here the different algal biomarkers which will be analyzed. Lines 21-26. **This has been added in a new section of the results (section 4.6, page 9, line 26 – page 10, line 4).** Please specify if the GC injections were made in split or splitless mode. **Injections were made in splitless mode. This has been added to section 3.2 (page 6, line 27).**

 Lines 1-10. A short introduction of 5- and 6-methyl brGDGTs and the related indices should be added. Were some samples injected in duplicate/ triplicate? *What is the analytical uncertainty on the MBT, CBT and IR indices?* **We have added an introduction to 5- and 6-methyl brGDGTs (Page 3, Lines 3-14). In addition, section 4.5 "BrGDGT isomer ratios" also discusses this topic (page 9, lines 19-25). A subset of the samples (n=32) were analyzed in duplicate. The analytical uncertainty from these measurements has been added to this section (page 7, lines 18-20).**

Lines 20-25. The relative abundances of brGDGTs in all the samples (soils, SPM, lake sediments) should be given in a supplementary table. **All Basin Pond data presented in this manuscript will be archived at the NOAA Paleoclimate Data Center when the manuscript is accepted for publication.**

Acyclic and cyclic brGDGTs (Ia, Ib and Ic; IIa, IIb and IIc; IIIa, IIIb and IIIc) cannot be distinguished in Fig. 2. The figure should be modified to take this comment into account. **We have noted relative abundances of the b and c compounds (although they are very small) on Page 8 lines 13-14. As this manuscript does not explicitly discuss the differences in abundances of**

the a b, and c, we chose not to display them in Figure 2 to make the figure more readable. Interested parties can access the a, b, and c data in the data tables available upon publication.

The different types of green are difficult to distinguish in Fig. 2. Please choose more contrasting colours. **We have updated this figure to use more contrasting colors.**

The brGDGT distributions are not consistent through the four collection periods: the relative abundance of GDGT I decreases from June 2014 to January 2015, in contrast with brGDGT III. **We originally made this observation on Page 6, lines 12–14. However, in June and July 2014, Group I brGDGTs were the most abundant, whereas in September 2014 and January 2015, reductions in Group I brGDGTs were accompanied by increases in Group III brGDGTs. We removed the sentence about distributions being consistent through time, and added additional sentences that accurately describe the changes (Page 8, Lines 14-19).**

Lines 4-5. These sentences are redundant with those in lines 1-3. **This paragraph was rearranged for clarity and to reduce redundancy. However, we note that we separately discuss brGDGT fluxes and brGDGT distributions in this paragraph (which now appears at page 8, line 20- Page 9, Line 2); while it may seem like there is redundancy, we find upon careful reading of these lines that all of the information presented is new and necessary.**

Lines 24. These concentrations are not present either in a Fig. or a Table. Please provide the bulk data as Supplementary material. **All Basin Pond data will be archived at the NOAA Paleoclimate Data Center when the manuscript is accepted for publication.**

Lines 27-29. These multidecadal events are difficult to distinguish. **Thank you - to aid the reader in distinguishing these events, we have listed the specific timing of the multidecadal events in the manuscript (Page 9, Lines 13-16).**

Line 1. Please add also the recent paper by Russell et al. (2018) to the list of brGDGT lacustrine calibration. **In the original manuscript, the first sentences of the paragraph were discussing older lakes calibrations. The revised manuscript has an entirely new section (section 5.2) discussing the Russell et al. (2018) calibration for African lakes, as well as the Dang et al. (2018) calibration for Chinese lakes.**

Lines 9-13. The different arguments provided here are not convincing. The recent lacustrine calibration by Russell et al. (2018) could be applied to the NE US lakes, with all the caution needed (not the same region, difference in terms of stratification/mixing etc.). This is indeed the only calibration based on the recent analytical method proposed by Hopmans et al. (2016) allowing the separation of 5- and 6-methyl brGDGTs. Furthermore, as shown in Supp. information, the temperature variations inferred from the Russell et al. calibration are similar to those derived from the calibration by Sun et al. (2011). Therefore, the different calibrations provide different absolute temperatures (still very close, ca. 1 °C difference) but similar trends.

**We have made major revisions to the Discussion, including adding a new section (5.2) and presenting our data in Figure 7 with the Russell et al. 2018 African Lakes calibration and the Dang et al. 2018 Chinese Lakes calibration, as these are the only calibrations based on the analytical method used in this study. In order to highlight the caution needed when interpreting these results, we have also added a ternary diagram to this section (Figure 8) as well as a new paragraph of information (page 12, line 21 – page 13, line 2), which shows that brGDGT distributions of the samples in this study fall well outside those of the African Lake and Chinese Lake samples. This means the brGDGT producers in Basin Pond, and their sensitivity to temperature, are potentially different from those in the lakes studied by Russell et al. 2018 and Dang et al. 2018. We therefore wish to emphasize that presenting the data in this way requires caution, as it invites a suite of misinterpretations for readers who are not familiar with the proxy. Because of this, we decided to present our data as MBT'$_{5ME}$ values in Figures 9 and 10, and indicated with words and colors in these figures that higher MBT'$_{5ME}$ values indicate higher temperatures, and vice versa.**

Line 18. Only group I and III brGDGTs can be distinguished, not the individual compounds. **We have corrected the text to reflect this. We note that the b and c compounds make up <2% of the relative abundance in our samples. The full data will be available online with the NOAA Paleoclimate Data Center for interested parties.** CBT'5Me and MBT'5Me were inverted in Fig. 6. **Thank you - this has been fixed.**

Line 22. Lower instead of higher. **Thank you - this has been fixed.**

Lines 1-6. This is redundant with the sentences above. **Thank you – we have edited this paragraph to eliminate redundancy.**

Lines 6-10. In addition to the brGDGT distribution, the brGDGT concentrations in soils, lake and SPM should also be compared. **brGDGT fluxes are explicitly discussed with reference to the SPM samples (e.g. Section 4.2, Page 8, Lines 12 – Page 9, Line 2, Figure 3). brGDGT concentrations in soils (Section 4.1, Page 8, Lines 7-10) and sediments (sections 4.3-4.4, Page 9, Lines 3-17) are also noted. These quantities do not have the same units and are thus not directly comparable. Furthermore, brGDGT concentrations have a wide range of values in the literature, making it difficult to interpret how the soil, sediment, and SPM concentrations compare and how they represent relevant environmental parameters. However, we have added in text stating that brGDGT concentrations in the Basin Pond sediment record do not correlate with MBT'5Me values, indicating that they are decoupled (Page 13, Lines 7-9).**

Lines 11-12. I would be very cautious about the MBT'5Me/temperature local calibration, as it is based on only 4 points. **We agree - this reasoning is why we put it in the supplement and noted that future work should address this promising but presently thin calibration.**

Lines 13-14. Where are the water temperature data? **Water temperature data are presented in Figure 4(a), and we have added a citation (Frost, 2005) which details the yearly cycle of water temperatures as a function of depth in Basin Pond.**

Lines 25-26. Such a calibration should have been developed in the present study and then applied to the downcore brGDGT reconstruction. **We have removed this sentence. Because of the limited number of SPM samples we were able to collect for this study, it is not a calibration experiment and should not be considered as such. Instead it is a detailed assessment of seasonal production biases coupled with the downcore application of a powerful emerging paleotemperature proxy.**

Line 31. These concentrations should be provided as Supp. Material. **These concentrations are included in the data tables that we will archive at the NOAA Paleoclimate Data Center upon publication.**

Line 33. Please remove the last sentence which is not useful. **This sentence has been removed.**

Line 9. Please be more explicit about similarities and differences between pollen and MBT'5Me reconstructions. **Text further discussing the differences and similarities, as well as expected differences in proxies, has been added in Section 5.3 (Page 13, Lines 16-24).**

Line 16. Please provide some references here. Are hydrogen isotopes from leaf waxes not mainly used are hydrological proxies? **Hydrogen isotopes of leaf waxes can reflect changes in temperature (e.g. high-latitude sites) changes in hydrological processes (e.g. tropical sites), changes in the dominant moisture source, or a mix of these and other processes, and interpretation is dependent on site-specific parameters. This is briefly discussed in the text (Page 2, Lines 15-17) but the details of these processes are beyond the scope of this study. We provide the $\delta$D record from Gao et al. 2017 with its published interpretation (as a temperature proxy) because it is one of the few available records from New England.**

Lines 16-21. This paragraph should be more developed. How are temperatures reconstructed from delta D of leaf waxes? **We have attempted to clarify the manuscript text that Gao et al. (2017) interpret their $\delta$D record as a temperature proxy (at page 13, lines 25-26). We provide these data simply for comparison to our own record. We do not wish to question the published interpretation of Gao et al. (2017) of $\delta$D as a temperature proxy.**

Line 28. Where are this information derived from? Fig. 7d? **This information is derived from comparison between Figure 9g and Figure 9d–f. This has been updated in the text (Page 14, Lines 12-13).**

Lines 29-32. The conclusion about the predominant human impact on fires remains speculative and is difficult to apprehend. **We agree. The speculative sentence about the causes of fire changing through time has been removed.**

Lines 13-15. The similarities between Basin Pond reconstruction and other northern

hemisphere reconstructions are difficult to visualize. **We have made the text more explicit about the similarities and differences between the Basin Pond reconstruction and northern hemisphere reconstructions (Page 14, Lines 24-29).**

Line 2. What do the authors mean by "not a strong cross correlation? *Please provide r and p values.* **Fixed. r and p values have been added to text (Page 15, Line 21; Page 16, Line 5).**

Lines 13-14. This trend is difficult to visualize. **We are unsure exactly which trend the referee is referring to but all the trends discussed are shown in Figure 10. Additional text was added to Section 5.4 to guide the reader to see all the trends we describe (Page 14, Lines 24-29; Page 15, Line 8). We have also clarified this paragraph by removing superfluous information regarding other trends that are not relevant to the present manuscript.**

Lines 17-18. What is the interest of presenting local data since the authors consider them as inaccurate? I would only present statewide trends. **We agree. We have adjusted the text(Page 16, Lines 11-17) and Figure 11 to remove local data and to only show statewide averages.**

Lines 24-27. This is a little too short. What are exactly the mechanisms which could explain the lower MBT'5Me values in surficial sediments? **We have added discussion, which lists possible mechanisms to explain this trend without speculating too much about our own data (Page 17, Lines 4-7). Furthermore, we have cited other studies (e.g. Tierney et al., 2012; Sinninghe Damste et al., 2012) that show a similar trend in the uppermost surface sediments, and discussed the fact that the responsible mechanisms are not known (Page 17 Line 30 – Page 18, Line 2).**
Some of these "mechanisms" may be lake-/region-dependent. **We agree.**

Lines 4-16 / Fig. 9a. The different algal biomarkers are difficult to distinguish in Fig. 9a. Please use contrasting colours for each biomarker. **We agree with the referee that the way it was arranged made it difficult to distinguish the colors and link them to the respective biomarkers. We have edited this figure to make it more reader-friendly in response to the referee's request. However, in order to present data in a color-blind friendly manner, we decided to stick with shades of gray. After uploading this figure to an online color blindness simulator, we have confirmed that it is viewable in its current format to all readers.**

Supplementary information

Line 17. Where is Table S1? **Table S1 is the first table in the supplement, but was mislabeled as S2. This has been fixed.**

Line 27. Where are the fall measurements? **The average measured temperatures for each of the sediment trap collection periods are listed in Table S2.**

A MBT'5Me-temperature calibration should have been developed in the present study. **We agree with several of your comments above that while this is a fantastic goal, we found that a correlation based on only four scattered points would be spurious (see following comment). Nevertheless, we do present a preliminary calibration in the supplement and supplementary figure S1 plots the Basin Pond brGDGT record on this calibration.**

The correlation presented in Fig. S1, based on only 4 scattered points, is not reliable. We agree. **This was our reasoning as to why we chose to include it in the supplement and not actually apply it to the data in the main text.**

[revised manuscript text omitted]